# Interaction of genetic variants activates latent metabolic pathways in yeast

Srijith Sasikumar ●[1,2,3], Shannara Taylor Parkins[4], Suresh Sudarsan[4] & Himanshu Sinha ●[1,2,3] ✉

Genetic interactions are fundamental to the architecture of complex traits, yet the molecular mechanisms by which variant combinations influence cellular pathways remain poorly understood. Here, we answer the question of whether interactions between genetic variants can activate unique pathways and if such pathways can be targeted to modulate phenotypic outcomes. The model organism *Saccharomyces cerevisiae* was used to dissect how two causal SNPs, $MKT1^{89G}$ and $TAO3^{4477C}$, interact to modulate metabolic and phenotypic outcomes during sporulation. By integrating time-resolved transcriptomics, absolute proteomics, and targeted metabolomics in isogenic allele replacement yeast strains, we show that the combined presence of these SNPs uniquely activates the arginine biosynthesis pathway and suppresses ribosome biogenesis, reflecting a metabolic trade-off that enhances sporulation efficiency. Functional validation demonstrates that the arginine pathway is essential for mitochondrial activity and efficient sporulation only in the double-SNP background. Our findings show how genetic variant interactions can rewire core metabolic networks, providing a mechanistic framework for understanding polygenic trait regulation and the emergence of additive effects in complex traits.

Complex traits arise from intricate interactions between causal genetic loci, the broader genetic background, and environmental influences[1,2]. These loci can influence traits through additive effects or non-additive interactions, such as dominance and epistasis. Several studies have focused on mapping genetic interactions in model organisms like yeast[3–8] and human diseases[9–13]. A global genetic interaction network for yeast employing reverse genetic screens on essential and non-essential genes revealed numerous genetic interactions, with many interacting genes belonging to the same functional pathways, forming highly organized and structured genetic networks[6]. Similarly, CRISPR knockout screens have identified genes essential for cell viability in cancer and pluripotent stem cells in human cell lines[14]. While most studies have focused on gene deletions or knockouts to identify genetic networks, a few have used biparental yeast populations to find

genetic networks among gene variants[7,15]. These studies identified variant-specific gene networks that had environment-sensitive genetic interactions, and incorporating such genetic interactions into predictive models of phenotypic variation significantly improved their accuracy.

While the studies mentioned above highlight the prevalence of genetic interactions in shaping complex traits, a crucial gap remains in understanding how these interactions function at the molecular level. Specifically, how do genetic variants interact to modulate molecular pathways and contribute to phenotypic variation? Addressing this question is essential for identifying novel functional targets that can be leveraged to modulate complex traits, particularly in diseases influenced by multiple interacting variants[16,17]. In this study, we sought to test the hypothesis: do interacting SNPs function independently

[1]Systems Genetics Lab, Department of Biotechnology, Bhupat and Jyoti Mehta School of Biosciences, Indian Institute of Technology Madras, Chennai, India. [2]Centre for Integrative Biology and Systems Medicine (IBSE), Indian Institute of Technology Madras, Chennai, India. [3]Wadhwani School of Data Science and Artificial Intelligence (WSAI), Indian Institute of Technology Madras, Chennai, India. [4]The Novo Nordisk Foundation Centre for Biosustainability, Technical University of Denmark, Lyngby, Denmark. ✉e-mail: sinha@iitm.ac.in

through their respective functional networks, or do they activate latent molecular pathways when combined (Fig. 1A)?

There are several challenges in studying genetic interactions between SNPs, particularly in complex traits. First, genetic background can confound the effect of SNP interactions[5,18,19]. Second, while gene expression has been widely studied, its correlation with protein levels is often limited[20–22]. Third, biological processes and the intermediate phenotypes underpinning these processes are dynamic[23].

To overcome these challenges, we employed *S. cerevisiae*, a yeast model system, where one can use isogenic allele replacement yeast strains to study various complex traits and identify the phenotypic and molecular effects of genetic variants at the SNP level[3,24,25]. Gene expression changes alone may not fully explain phenotypic outcomes, as protein expression and post-translational modifications frequently play critical roles. A systems genetics approach of integrating multi-omics data, such as transcriptomics, proteomics, and metabolomics, offers a framework for overcoming the challenge of understanding the complex interplay among these intermediate phenotypes and their impact on biological processes[26,27]. In recent studies, the integration of multi-omics information has guided targeted therapies in discovering biomarkers for cancers and other complex diseases[28–34]. Furthermore, studies in model organisms[35–41] and humans[42] have demonstrated that gene and protein expression variations are highly context-specific, varying by developmental stage. Despite significant advances, most studies investigating the molecular basis of genetic interactions in yeast have focused on gene expression at a single time point, including our previous study[43,44]. Therefore, understanding the temporal phase during which causal genetic variants exert their molecular effects is critical for understanding genotype-phenotype relationships[37,40,41].

Here, we designed a study to systematically investigate the genetic interaction between two key quantitative trait loci (QTLs) in yeast sporulation, a developmental process. Using isogenic yeast strains carrying distinct SNPs, individually and in combination, we integrated multi-omics data to capture gene and protein expression and key metabolite level changes at multiple time points during sporulation (Fig. 1B). This allowed us to compare the effects of individual SNPs with their combined effects, revealing how SNP interactions modulate molecular pathways over time. Our results showed that the combination of SNPs activated a latent metabolic pathway, different from those activated by each SNP independently. Our findings have broader implications for understanding polygenic traits and diseases, where multiple SNPs contribute additively to phenotypic variation. We provide critical insights into the regulatory mechanisms underlying complex traits by dissecting how causal SNPs interact at the molecular level. This knowledge can inform the development of therapeutic strategies for diseases driven by polygenic interactions, offering new avenues for targeted interventions.

## Results

### Role of *MKT1^89G* and *TAO3^4477C* SNPs in sporulation efficiency variation

Sporulation is a developmentally regulated metabolic process in yeast. In response to nutrient limitations, such as nitrogen starvation and the availability of a non-fermentable carbon source like acetate, diploid yeast cells undergo meiosis, ultimately producing four haploid spores[45]. Sporulation efficiency, defined as the rate of spore formation, is a quantitative trait in yeast and is well-studied[35,36,46–48]. A previous study by Deutschbauer and Davis[25] identified *MKT1^89G* and *TAO3^4477C* SNPs from the high sporulating strain, SK1, as causal for increasing sporulation efficiency of the low sporulating strain, S288c. From the comprehensive yeast genomics dataset of 3034 strains[49], we found that the *MKT1^89G* SNP has an allele frequency of 0.9955, indicating that it is a common variant in *S. cerevisiae*. In contrast, the *TAO3^4477C* SNP has an allele frequency of $4.952 \times 10^{-4}$, highlighting that it is an ultra-rare

allele in the population. As these are independent SNPs, we expect around 0.0493% of the population to carry both alleles in combination.

To study the phenotypic effects of these two SNPs, we used a panel of isogenic strains in the S288c background generated by swapping specific S288c nucleotides with their SK1 allele counterparts, resulting in four diploid strains: wildtype S288c, hereafter referred to as SS strain, MM strain with the *MKT1^89G* SNP, TT strain with the *TAO3^4477C* SNP, and MMTT strain harboring both *MKT1^89G* and *TAO3^4477C* SNPs in combination (Fig. 2A). Under the conditions described[25], we first measured the sporulation efficiencies of these allele replacement strains to assess the effects of SNPs on sporulation efficiency variation. The MM (39.41 ± 2.42%), TT (37.42 ± 1.81%), MMTT (75.42 ± 3.68%), and MmTt (71.29 ± 0.52%) allele replacement strains, along with the wild-type SK1 strain (90.13 ± 1.78%), exhibited higher sporulation efficiency after 48 h in sporulation medium with acetate as sole carbon source compared to the S288c strain (SS: 7.00 ± 1.54%, Fig. 2B), consistent with previous reports[25,40,41]. We confirmed that the sporulation efficiency phenotype of the MMTT was additive, representing the sum of individual sporulation efficiencies of the MM and TT strains. Furthermore, we observed that the *MKT1^89G* and *TAO3^4477C* alleles were dominant over the S288c alleles *MKT1^89A* and *TAO3^4477G* (Fig. 2B).

Sporulation in yeast is induced when strains are grown in an acetate medium, the sole carbon source. We hypothesized that increased sporulation efficiency of MM, TT, and MMTT strains could be due to altered acetate uptake, as previous studies have shown activation of key metabolic pathways like nitrogen metabolism, TCA cycle, and gluconeogenesis associated with the SNPs involved[40,41]. To test this, we measured extracellular acetate levels over time. MMTT strain showed a sharp decline in acetate compared to SS, MM, and TT strains, which were similar at 2 h but diverged later (Fig. 2C). Further, we observed that the MM and TT strains outperformed SS in utilization, particularly after 8 h (Fig. 2C). Intracellular acetate analysis revealed rapid consumption in MMTT within 8 h, followed by gradual accumulation up to 24 h (Fig. 2D), suggesting activation of downstream metabolism during the early stages of sporulation. The TT strain showed a biphasic usage with peaks at 2 h 30 min and 8–12 h. The SS and MM strains had similar trends with an early dip, followed by an accumulation till 12 h and then a decline.

These observations suggested that the enhanced sporulation efficiency observed in the MMTT strain could be driven by more efficient acetate uptake and utilization during the early time points (0–8 h). The distinct temporal patterns of intracellular and extracellular acetate across strains suggest that the SNP-associated changes can confer metabolic plasticity across strains. This plasticity is likely orchestrated by coordinated changes at the transcriptomic, proteomic, and intracellular metabolite levels, enabling differential sporulation efficiency across strains.

### Genetic interaction reshuffles amino acid metabolism and ribosomal pathways

In previous studies from our lab, we investigated the functional and molecular effects of the *MKT1^89G* and *TAO3^4477C* SNPs using global temporal microarray analyses, with samples collected throughout sporulation up to 8 h 30 min, with denser sampling during the early stages[40,41]. The choice of this logarithmic time series instead of a linear time series was to capture the rapid changes occurring in early response genes that have causal effects on the phenotype[40,41,50]. These studies revealed that *MKT1* and *TAO3* play critical roles early in sporulation. The *MKT1^89G* allele enhanced sporulation efficiency by activating genes such as *RTG1/3*, which are involved in mitochondrial retrograde signaling, and *DAL82*, associated with nitrogen starvation[40]. In contrast, the *TAO3^4477C* allele improved sporulation efficiency through the activation of genes like *ERT1*, linked to the TCA cycle, and *PIP2*, involved in gluconeogenesis, compared to when these SNPs were absent[41].

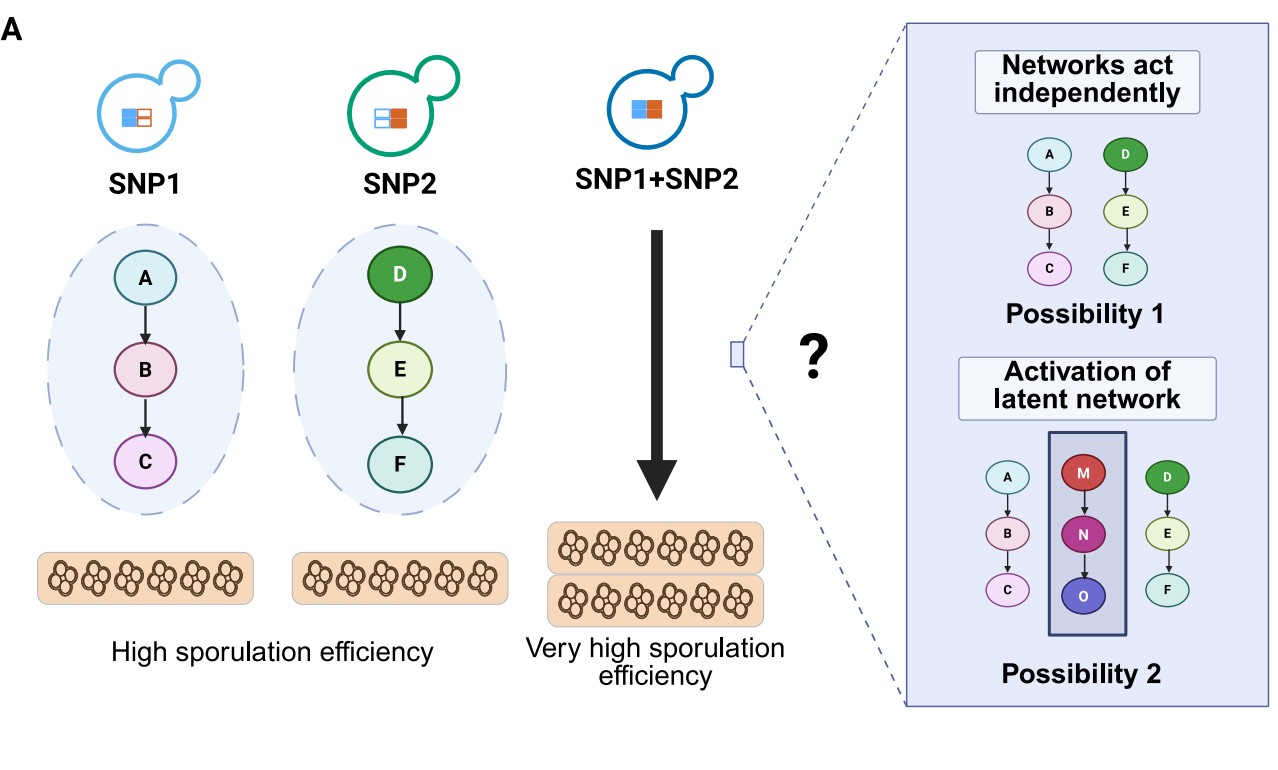

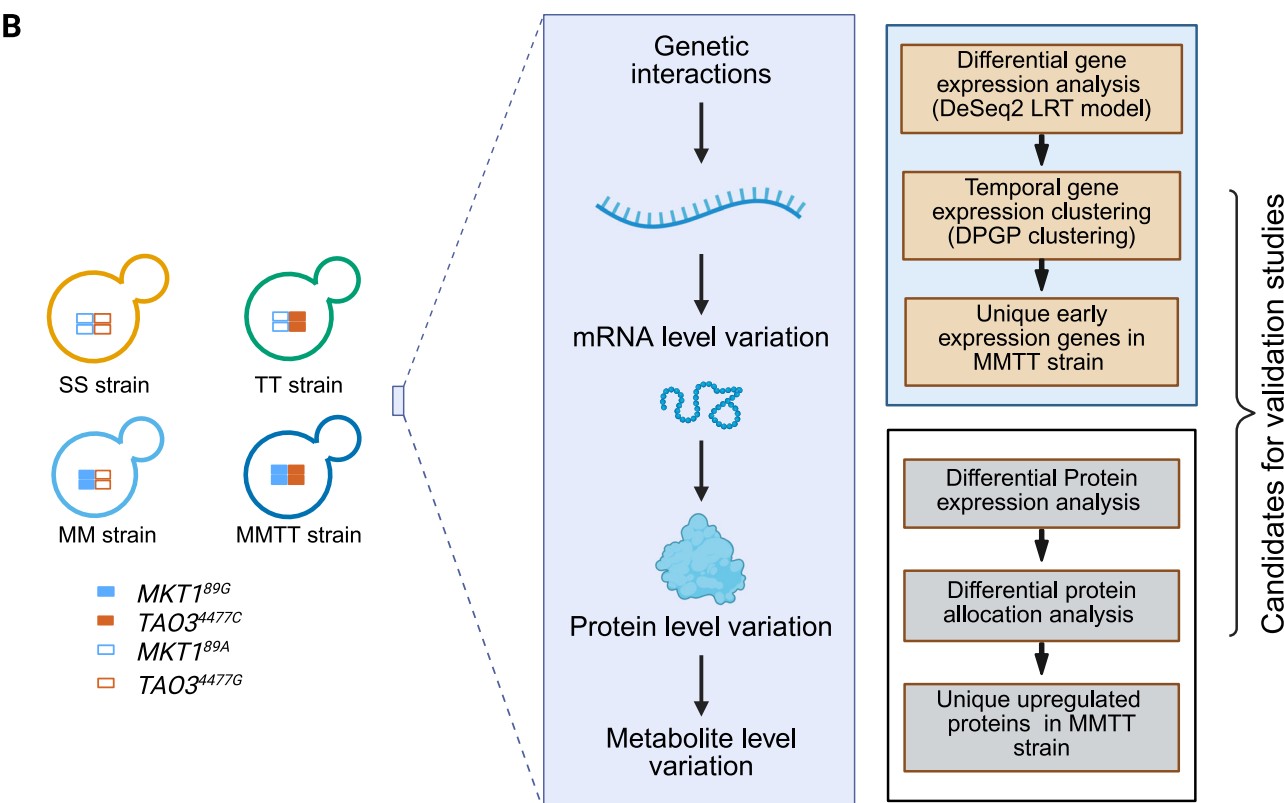

**Fig. 1 | Overview of the model under study. A** Illustration of the proposed molecular mechanism involving *MKT1*[89G] and *TAO3*[4477C]. This panel highlights the potential interaction between the gene and protein expression networks of *MKT1*[89G] and *TAO3*[4477C], contributing to the observed additive effect on higher sporulation efficiency. **B** Schematic representation of the study design for temporal transcriptomics and proteomics analysis of yeast strains in sporulation medium. This panel outlines the experimental approach to identify the molecular mechanisms underlying the phenotypic additivity observed in the yeast strains. Created in BioRender. Sinha, H. (2025) https://BioRender.com/07eizi3.

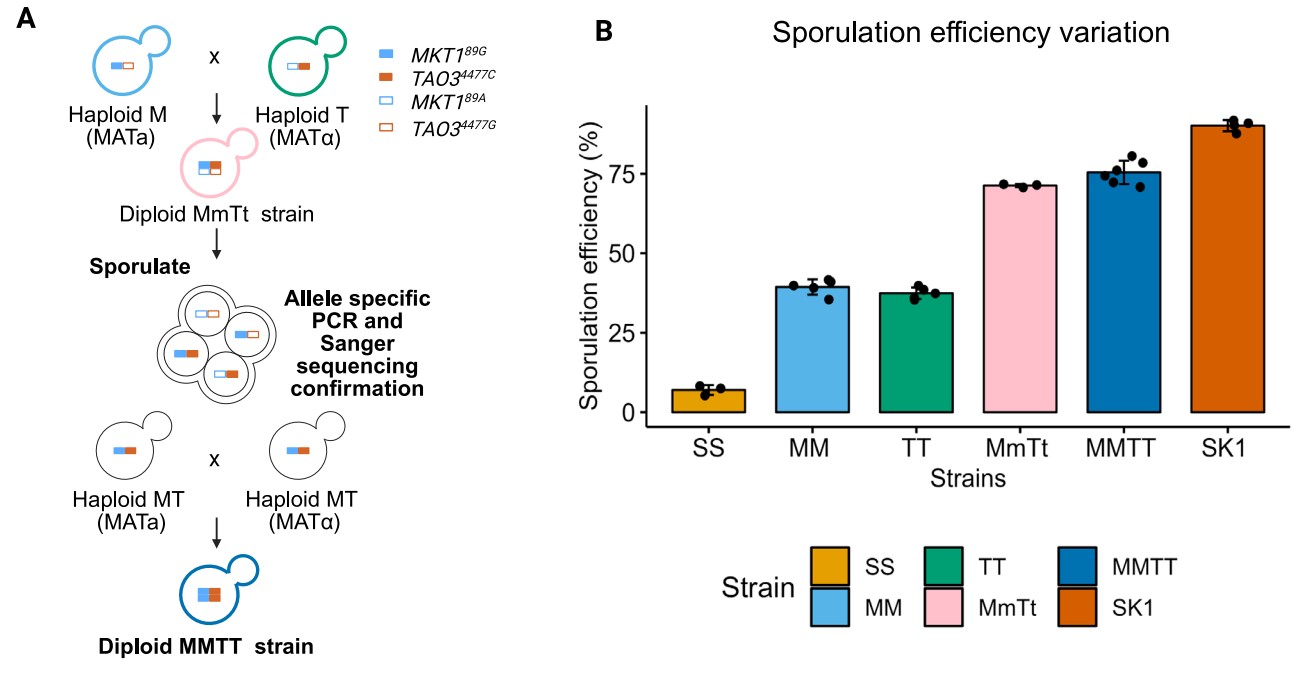

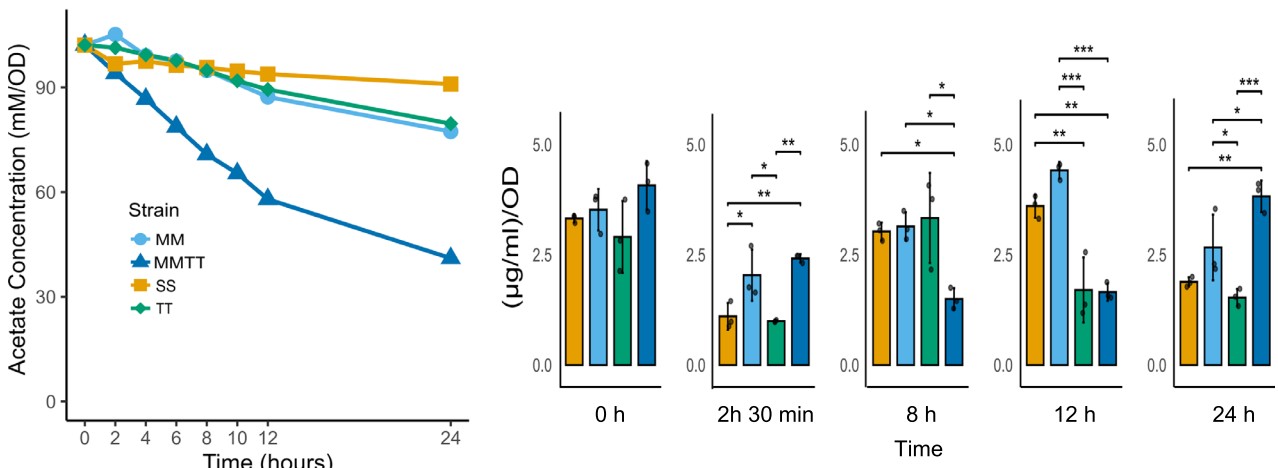

**Fig. 2 | Additive effect of *MKT1^89G* and *TAO3^4477C* on sporulation efficiency variation. A** Schematic representation of the steps involved in generating the MMTT strain, highlighting the introduction of the *MKT1^89G* and *TAO3^4477C* alleles. **B** Sporulation efficiency of the SS, MM, TT, MMTT, MmTt (heterozygous), and SK1 strains was measured after 48 h in the sporulation medium. The bar graph represents the percentage of sporulating cells (dyads, triads, and tetrads), demonstrating the additive effect of the *MKT1* and *TAO3* alleles. Error bars represent mean ± SD from at least three biological replicates. Individual data points are shown as dots. **C** Mean values of extracellular acetate levels in the sporulation medium of the SS, MM, TT, and MMTT strains at different time points during

sporulation ($N = 2$). **D** Intracellular acetate concentrations were measured across four yeast strains (SS, MM, TT, MMTT) at five time points (0 h, 2 h 30 min, 5 h 40 min, 12 h, and 24 h) in sporulation medium. Error bars represent mean ± SD from three biological replicates. Individual data points are shown as dots. Statistical significance was assessed using one-way ANOVA, followed by Tukey's HSD post-hoc test for each time point independently. Significance levels are indicated as follows: ****$p < 0.0001$, ***$p < 0.001$, **$p < 0.01$, *$p < 0.05$. Exact adjusted *p*-values are provided in the Source data file. Source data are provided as a Source Data file for (**B**–**D**). Created in BioRender. Sinha, H. (2025) https://BioRender.com/07eizi3.

To characterize the effect of *MKT1^89G* and *TAO3^4477C* SNPs in the combination on the global gene expression variation, we performed RNAseq for SS strain and MMTT strain throughout sporulation with denser sampling during the early phase at 0 h, 30 min, 45 min, 1 h 10 min, 1 h 40 min, 2 h 30 min, 3 h 50 min, 5 h 40 min till 8 h 30 min (Fig. 3A). These time points were identical to the expression analysis study done for single SNP strains, MM and TT[40,41], which allowed us to make comparisons of differentially expressed genes between

individual SNPs and their combinations to identify causal mechanisms activated in response to genetic interactions.

Hierarchical clustering and Principal component analysis (PCA) of the gene expression data revealed a clear distinction between the SS and MMTT strains (Supplementary Fig. 1A, B). The analysis of differentially expressed genes at each time point against the initial time point between the SS and MMTT strains revealed substantial differences in gene expression patterns between the two strains.

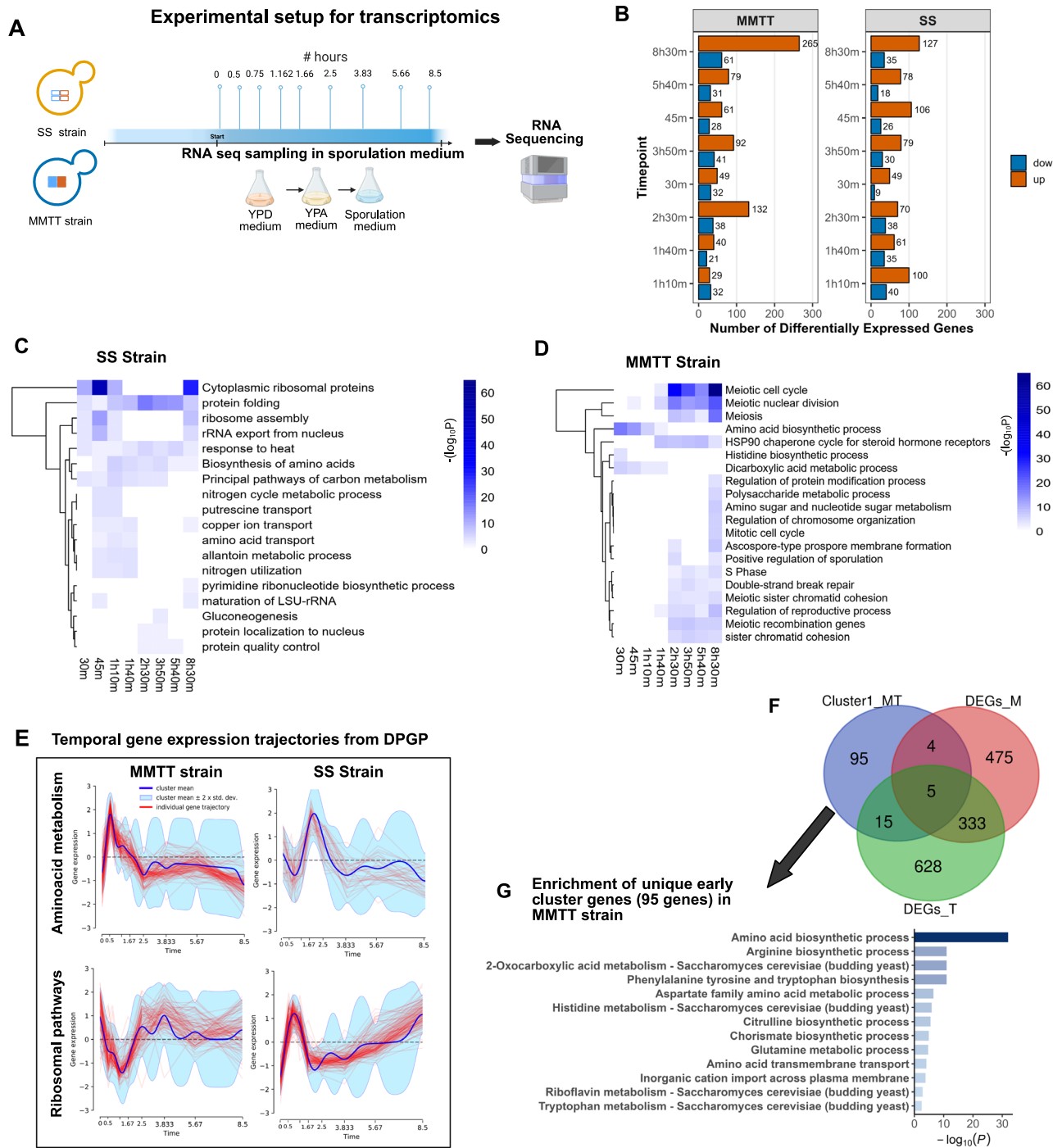

**Fig. 3 | Temporal reshuffling of amino acid metabolism and ribosomal pathways in the presence of *MKT1^89G* and *TAO3^4477C* SNPs. A** Schematic representation of the design of the time-series RNA-seq experiment. This panel illustrates the overall experimental setup, including RNA sample collection from SS and MMTT strains at 9 time points during sporulation. **B** Bar plots showing the number of differentially expressed genes (DEGs) at each time point compared to 0 h in SS and MMTT strains. DEGs were identified using DESeq2 with Wald test (two-sided), applying an adjusted $p$-value < 0.05 (Benjamini–Hochberg correction) and |$\log_2$ fold change| > 2. **C, D** GO enrichment analysis of upregulated DEGs using Metascape for the SS strain (**C**) and MMTT strain (**D**) across all time points. Enrichment $p$-values were calculated using the cumulative hypergeometric test (one-sided) and adjusted for multiple comparisons using the Benjamini–Hochberg method

(Metascape). Heatmaps show $-\log_{10}(p\text{-value})$ for the top enriched GO terms. **E** Temporal gene expression trajectories of genes involved in amino acid metabolism and ribosomal pathways, derived from DPGP clustering. Shaded regions represent ±2 standard deviations (SD) around the mean expression trajectory. **F** Venn diagram comparing early clustered genes in MMTT (Cluster 1) with DEGs from MM and TT strains. **G** GO enrichment analysis of the 95 genes uniquely present in Cluster 1 of the MMTT strain using Metascape. Enrichment $p$-values were calculated using the cumulative hypergeometric test (one-sided) and adjusted using Benjamini–Hochberg correction (Metascape). Bar plot represents $-\log_{10}(\text{adjusted } p\text{-value})$. Source data are provided as a Source Data file for (**C, D**). Created in BioRender. Sinha, H. (2025) https://BioRender.com/07eizi3.

The differentially expressed genes at each time point for SS and MMTT strains are provided in Supplementary Data 1. A notable increase in DEGs was observed in the MMTT strain, particularly during the early phase (2 h 30 min), with 132 genes upregulated and 32 downregulated. This trend continued in the later phase (8 h 30 min), where 265 genes were upregulated, and 61 were downregulated (Fig. 3B). GO enrichment analysis on the upregulated and downregulated genes found that in the case of the SS strain, there was a consistent upregulation of pathways like protein folding, response to heat (stress response), and biosynthesis of amino acids across all time points (Fig. 3C) In contrast, the MMTT strain exhibited distinct pathway activations between early time points (30 min, 45 min, 1 h 10 min, 1 h 40 min) and middle to late time points (2 h 30 min, 3 h 50 min, 5 h 40 min, 8 h 30 min). Specifically, significant upregulation was observed in genes associated with the amino acid biosynthetic process, dicarboxylic acid metabolic process, and histidine biosynthetic process as early as 30 min into sporulation (Fig. 3D). These pathways remained upregulated during the early phase but not in the middle and late phases. Additionally, the enrichment of pathways related to meiotic nuclear division and the meiotic cell cycle began at 1 h 40 min in the MMTT strain, which was not observed in the S strain. Furthermore, genes involved in polysaccharide metabolic processes and ascospore-type pro-spore membrane formation were uniquely upregulated at 8 h 30 min in the MMTT strain, highlighting the onset of spore wall formation (Fig. 3D).

We also compared differential gene expression at each time point between the SS and MMTT strains. As expected, we identified a large number of differentially expressed genes in the MMTT strain, particularly at 2 h 30 min and 8 h 30 min (Supplementary Fig. 2, Supplementary Data 1). During the early stages of MMTT strain, upregulated genes were significantly enriched in pathways related to amino acid biosynthesis, including histidine metabolism and L-arginine biosynthetic processes. In the later stages, we observed enrichment of genes involved in the meiotic cell cycle and meiosis only in the MMTT strain (Supplementary Fig. 3).

In addition, we identified a set of genes that were differentially expressed as a function of the $MKT1^{89G}$ and $TAO3^{4477C}$ genotypes in combination over time. The DESeq2 LRT method[51] was applied with a full model that included genotype + time + genotype × time and a reduced model that included only genotype + time. This analysis identified 1080 differentially expressed genes with an adjusted p-value of 0.001 (Supplementary Data 2). These genes were primarily enriched in pathways related to the meiotic cell cycle, cytoplasmic ribosomal proteins, amino acid metabolic processes, and ribosome biogenesis (Supplementary Fig. 4A).

To understand the temporal trajectories that vary between the SS and MMTT strains, we performed clustering using the Dirichlet Gaussian process mixture model (DPGP)[52]. Based on recent studies that have established the effectiveness of the DPGP model in identifying co-regulated genes within transcriptomic datasets, we applied DPGP clustering independently to the expression values of these 1080 differentially expressed transcripts for both the SS and MMTT strains. In brief, DPGP clusters the data using the Dirichlet process while modeling the temporal dependencies with Gaussian processes for non-uniform time points. This analysis resulted in 17 clusters for the MMTT and 18 for the SS strain. The genes in each cluster for the two strains are detailed in Supplementary Data 3. The clusters were then analyzed for enrichment to specific biological processes using Metascape (Supplementary Fig. 4B, C), and the trajectories of each cluster are shown in Supplementary Fig. 5A, B. Given the importance of early response genes in determining sporulation efficiency variation, the GO enrichment for the early cluster genes was specifically examined. In the SS strain, clusters 5 and 9 (Supplementary Figs. 4B, 5A), which showed early expression trajectories, were enriched for ribosome-related pathways, with 49 and 121 genes, respectively (Fig. 3E). In the MMTT strain, cluster 5, enriched for ribosome-related pathways

(Supplementary Fig. 4C), exhibited the downregulation of genes during the early phase (Fig. 3E).

We also observed that cluster 1 in the MMTT strain, enriched for amino acid metabolism genes, showed an early expression trend (Fig. 3E). In comparison, these pathways in the SS strain displayed an initial downregulation followed by delayed upregulation (Fig. 3E). This pattern suggested a trade-off in the MMTT strain between the amino acid biosynthetic process and ribosomal pathways. Since ribosome biogenesis was an energy-intensive process, the MMTT strain appeared to downregulate ribosome-related genes and upregulate amino acid metabolism in response to sporulation conditions, possibly to enhance sporulation efficiency, a strategy not observed in the SS strain.

We then examined the active genes in the later stages of sporulation. As expected, clusters 2 and 3 in the MMTT strain, which showed increased and synchronized gene expression in the later phase, were enriched for meiosis-related pathways, cell cycle regulation, and pro-spore membrane formation (Supplementary Figs. 4C, 6). This differed from SS, where these genes did not display any specific temporal gene expression pattern (Supplementary Fig. 6). This finding demonstrated how the combination of $MKT1^{89G}$ and $TAO3^{4477C}$ reshuffled the trajectories of amino acid metabolism and ribosomal pathways to facilitate efficient sporulation in the MMTT strain.

Finally, we aimed to identify genes that exhibit early expression trends in the MMTT strain but do not show differential expression in the MM and TT strains. These genes are interesting because their expression likely results from genetic interactions of $MKT1^{89G}$ and $TAO3^{4477C}$ specific to the MMTT strain and could be causal for the sporulation efficiency variation. Comparison of the differentially expressed genes identified for the MM and TT strains, as reported in the original publications[40,41], with the 119 genes in cluster 1 of the MMTT strain revealed 95 genes unique to the MMTT strain, which were enriched in pathways related to amino acid biosynthesis, including phenylalanine, tyrosine, and tryptophan biosynthesis, 2-oxocarboxylic acid metabolism, and arginine biosynthesis (Fig. 3F, G). These findings underscore a distinct transcriptional response in the MMTT strain during sporulation to an immediate response to nitrogen starvation, driven by the genetic interactions between the $MKT1^{89G}$ and $TAO3^{4477C}$ SNPs, which enhanced sporulation efficiency.

## Limited but function-specific correlation between transcript and protein levels

Transcript abundance and protein level correlations are poorly correlated[20–22]. In this and previous studies[35,40,41], specific functional categories of gene sets were differentially and temporally regulated during sporulation. To elucidate if these SNP-specific gene expression variations had protein level changes and thereby directly affected phenotypic variation, we performed an absolute quantification of the yeast proteome in the sporulation medium for the four allele-specific strains (SS, MM, TT and MMTT). Since most of the SNP-specific gene expression changes were in early sporulation, we focused on elucidating the early-phase proteome dynamics during sporulation, as differential expression patterns observed during this stage indicate genotype-driven phenotypic changes.

Employing label-free absolute quantitative proteomics using data-independent acquisition and Total Protein Approach (TPA approach), we quantified protein concentrations (in fmol μg⁻¹) at 2 time points, once at an initial time point (0 h) and during the early phase of sporulation (2 h 30 min). This approach provided a direct and quantitative view of proteome composition across conditions and genotypes. The absolute quantification enabled biologically meaningful comparisons of protein mass fractions, which were critical for interpreting proteome allocation and cellular resource distribution as described in previous studies[20,53] (Fig. 4A, B, Supplementary Fig. 7). Pearson correlation of 2996 pairs of transcripts (TPM) and protein abundance values

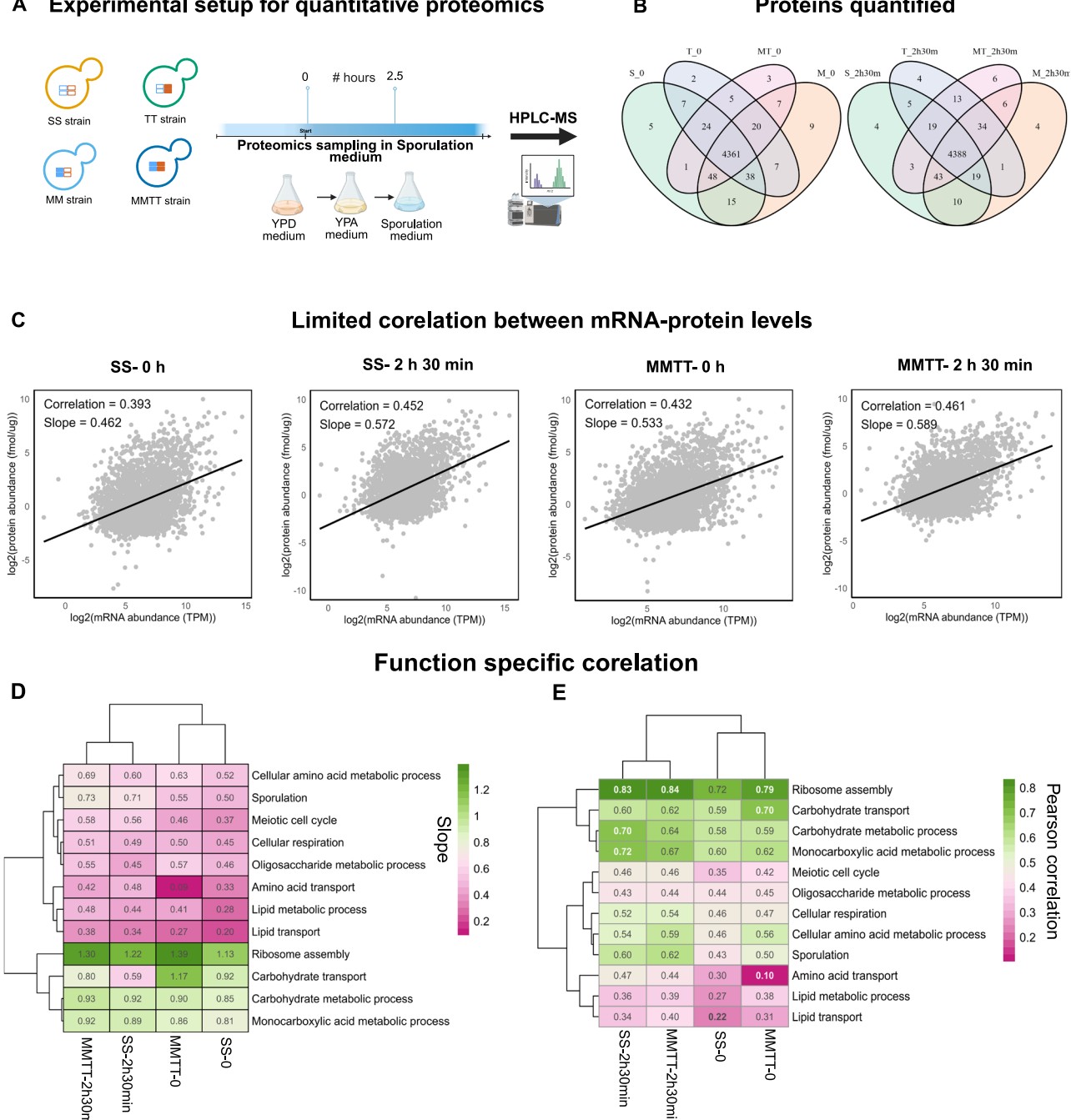

**Fig. 4 | Experimental setup for absolute quantitative proteomics. A** The S288c, MM, TT, and MMTT strains were grown in the sporulation medium, and sampling was done at the 0th hour and 2 h 30 min during sporulation. **B** Venn diagram showing overlap of proteins quantified at 0 h and 2 h 30 min for SS, MM, TT, and MMTT strains. **C** Correlation between mRNA and protein pairs at each time point (0 h and 2 h 30 min) for SS and MMTT strains. **D, E** The heatmap represents the correlation coefficients (**D**) and slope (**E**) of the integrated mRNA and protein abundance of various gene classes obtained for the GO-Slim mapper list. Created in BioRender. Sinha, H. (2025) https://BioRender.com/07eizi3.

(fmol μg⁻¹) for SS and MMTT strain showed a limited overall correlation between the datasets (correlation ranging between 0.393 and 0.461) and protein-mRNA slope represented as a regression line (depicts the relationship between the abundance of mRNA and the abundance of proteins) between 0.462 and 0.589 with $p$-value $< 2.2e^{-15}$ (Fig. 4C). These findings were consistent with previous findings in yeast[20,21] and human CD8⁺ T cells[54].

Given this limited correlation, we analyzed gene class-specific correlations. Genes were grouped based on the GO-slim mapper terms list, and their mRNA-protein slope and correlation were

calculated. We observed that the ribosomal assembly, carbohydrate transport, carbohydrate metabolic process, and monocarboxylic acid metabolic process showed a higher slope and correlation than other gene classes (Fig. 4D, E). Furthermore, our analysis revealed that the slope and correlation of gene classes specific for respiration, meiotic cell cycle, amino acid transport, cellular amino acid metabolic process and sporulation were comparatively higher for the early phase of sporulation (2 h 30 min) in MMTT strain compared to the 0 h of SS and MMTT and 2 h 30 min of SS strain (Fig. 4D, E). These observations highlighted that genes defined

by the time point display distinct mRNA-protein correlations during sporulation.

## Differential protein levels highly correlate with temporal transcriptome analyses

The presence of causal variants can influence the protein expression changes and differentially activate pathways in response to genetic interactions between variants. To test this hypothesis, we first calculated the mass fraction of each protein and then conducted pairwise differential abundance analysis as described previously[53] and briefly described in Methods. Differential protein abundance analysis revealed significant regulation of proteins across pairwise comparisons with the SS strain for respective time points (Supplementary Data 4, 5). We first verified whether *MKT1* and *TAO3* genes are differentially regulated at the protein level, given that no differences were observed at the gene expression level. Interestingly, we found that *MKT1* protein levels were nearly undetectable in both the SS and TT strains, which carry the *MKT1*[89A] variant. In contrast, *MKT1* protein was expressed in the MM and MMTT strains, both of which carry the *MKT1*[89G] variant. This indicates that the *MKT1* protein encoded by the *89A* variant likely has reduced stability or increased degradation compared to the *89G* variant. For *TAO3*, we did not observe any significant changes in either gene or protein expression across the SS, MM, TT, and MMTT strains. This suggests that the *TAO3*[4477C] variant may alter protein function rather than affecting expression levels or protein stability (Supplementary Fig. 8).

By studying the differentially expressed proteins, we found that, at the 0 h time point, we identified 123 proteins with significant regulation (105 up and 18 downregulated) when comparing the MM and SS strains. Similarly, between the TT and SS strains, 26 proteins were significantly regulated (20 up and 6 downregulated). Notably, a substantial number of genes (206) were deregulated when comparing the MMTT and SS strains, with 188 upregulated and 16 downregulated genes. Significant protein regulation was also evident during the early sporulation phase (2 h 30 min). In the MM strain, we identified 65 regulated proteins (42 up, 23 down); in the TT strain, 35 proteins (13 up, 22 down); and in the MMTT strain, 98 proteins (86 up, 12 down) compared to the SS strain (Fig. 5A). The upregulated proteins, both unique and shared among strains, are given in Table 1.

The gene ontology (GO)-term enrichment analysis revealed activated pathways or processes shared and unique between each SNP and their combination (Fig. 5B). Cellular respiration and mitochondrial translation processes were enriched during 0 h and 2 h 30 min sporulation (Fig. 5B) among the upregulated proteins in the MM and MMTT strains. This observation suggested a role for *MKT1*[89G] in enhancing cellular respiration and mitochondrial activity during the early phases of sporulation. In contrast, no significant enrichment of gene ontology terms was observed for the TT strain. Notably, we observed a unique enrichment of the arginine biosynthetic pathway exclusively for the MMTT strains at 0 h and 2 h 30 min (Fig. 5B, C). We also observed mitochondrial genome maintenance, respiratory chain complex 3 assembly, and regulation of mitochondrial gene expression to be enriched uniquely in the MMTT strain during the initiation of sporulation (Fig. 5B).

The temporal changes in protein abundances were examined by comparing the 0 h and 2 h 30 min time points across all four strains (Supplementary Fig. 9A). The number of differentially expressed proteins at the 2 h 30 min time point, in comparison to the 0 h time point, showed that 7 proteins were found to be upregulated in all strains at the 2 h 30 min time point, including Ady2 (required for acetate utilization during sporulation)[55], Ino1 (involved in myo-inositol biosynthesis and required for sporulation)[56], Cit3 (involved in the tricarboxylic acid cycle)[57], Ena1 (expressed in response to glucose starvation)[58], Pho89 (involved in phosphate metabolism)[59], Leu1 (involved in leucine biosynthesis)[60], Oac1 (oxaloacetate carrier)[61], and Nce103 (necessary

for the formation of bicarbonate, a step required for sporulation that increases the pH of the sporulation medium[62]; Supplementary Fig. 6B, Supplementary Data 6). Analysis of the downregulated proteins identified that in the MMTT strain specifically, several ribosomal proteins (Rtc6, Rpl22a, Rpl14b, Rpl4b, Rpl13a) were downregulated. This complemented our gene expression analysis, indicating that the MMTT strain reduced energy-intensive processes like ribosome biogenesis to promote sporulation.

## Genetic interaction increases proteome allocation to arginine biosynthesis and mitochondrial respiration

Further, we investigated how proteome resources were reallocated in response to the causal variants and temporal progression in the sporulation medium. This approach provides a holistic view of how yeast strains adapt to nutrient-limited conditions by reallocating their cellular resources and helping to identify key proteomic adjustments that contribute to efficient sporulation. For this, we calculated the summed mass allocations of a group of proteins belonging to a particular GO term obtained from the yeast GO-slim mapper process terms dataset[53]. We then compared the SS and allele replacement strains (MM, TT, and MMTT) at each time point. We focused on GO terms that exhibited a significant *p*-value for at least one of the comparisons independently for each time point. We observed that 17 GO mapper processes were reallocated at 0 h in response to causal variants (Supplementary Fig. 10A). At the same time, a substantial reallocation of proteins occurred during the 2 h 30 min time, with 36 GO mapper processes being significantly reallocated (Supplementary Fig. 10B). At 0 h, causal variants led to increased protein allocation to cellular respiration, as yeast cells had already been exposed to a pre-sporulation medium with acetate as the sole carbon source. We also noted a significant change in protein allocation to mitochondrial translation when the *MKT1*[89G] variant was present. After 2 h 30 min into sporulation, MM and MMTT strains showed significant enrichment for mitochondrial translation. Additionally, we found a unique enrichment of mitochondrial organization and amino acid transport in the MMTT strain.

As we identified an enrichment of upregulated proteins involved in the arginine biosynthetic process using differential protein abundance analysis, we wanted to examine how proteins are allocated to the arginine biosynthetic pathway. We found that the allocation to arginine metabolism increases during sporulation in all strains. Specifically, we found that the protein allocation to arginine metabolism in the MMTT strain was higher than in the SS, MM and TT strains during the 2 h 30 min period (Fig. 5D). We also observed that the protein is reallocated to amino acid metabolism from ribosomes and glycolysis during sporulation in a genotype-specific manner, suggesting yeast cells in nutrient-limited conditions redirect their protein resources towards amino acid metabolism rather than ribosomes, translation (Table 2), and glycolysis (Supplementary Fig. 11). This finding aligned with previous studies[53], demonstrating that yeast reallocated proteins to ribosomes in rich conditions, enhancing growth rates.

Further, to validate our findings, we assessed the cellular energy status by analyzing the intracellular ATP levels for all strains during sporulation at multiple time points. We observed that all the strains had a sharp decline in ATP levels by 2 h 30 min, with the MMTT strain showing a complete recovery of the initial ATP concentration during 8 h, while other strains showed partial recovery (Supplementary Fig. 12). This enhanced recovery highlights the more efficient respiration during sporulation in the MMTT strain. Further to understand the increased ATP levels seen in the SS strain, we analyzed the protein allocated to the glycolytic pathway, a key pathway for ATP production. At 2 h 30 min, the SS strain maintained a higher proportion of its proteome allocated to glycolytic enzymes (-14%), while MM, TT, and MMTT strains showed reduced allocation (-11–12%, Supplementary Fig. 11). This suggested that the SS strain retained a stronger glycolytic capacity, potentially supporting ATP

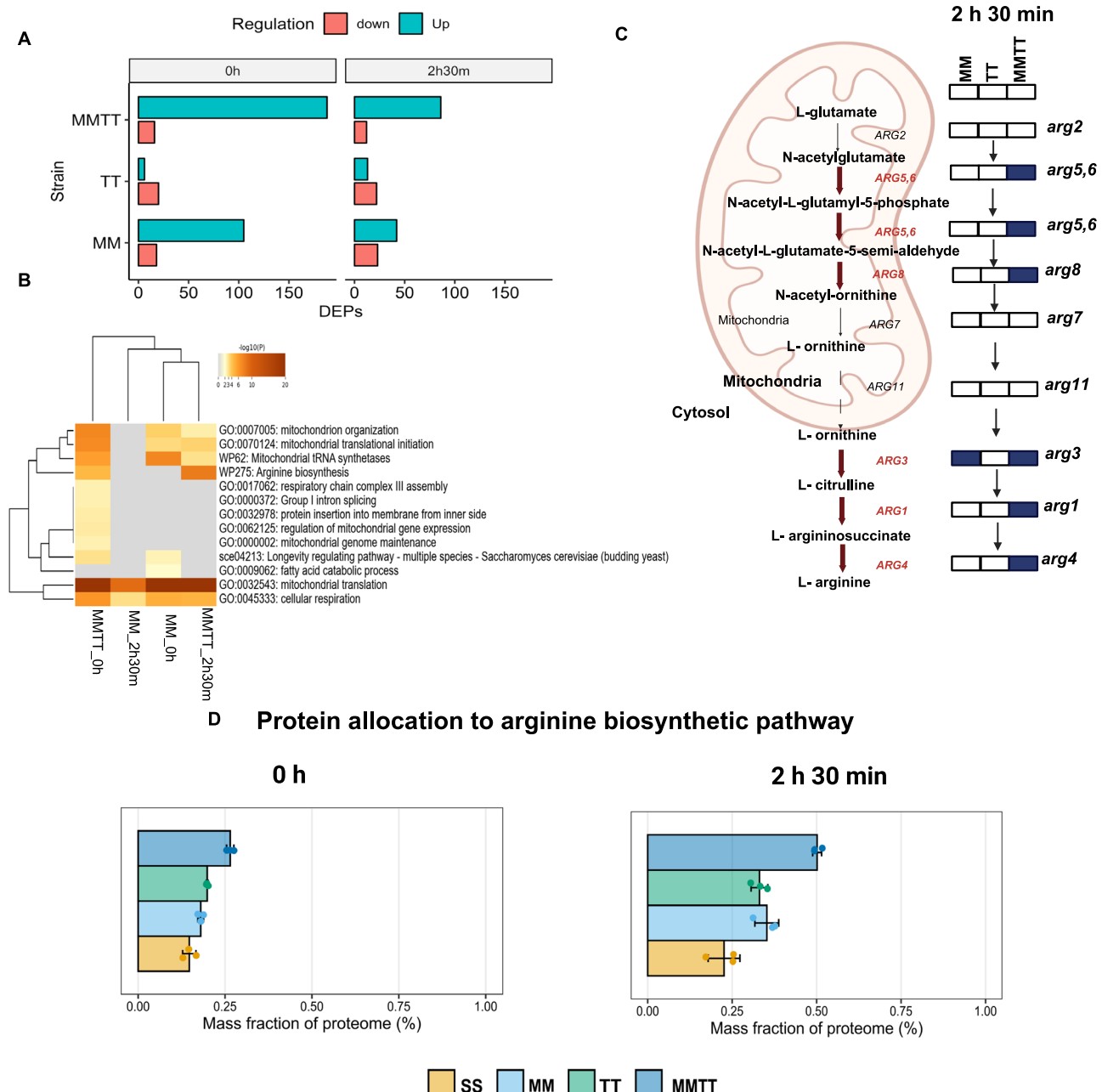

**Fig. 5 | Differential enrichment of the arginine biosynthetic pathway when *MKT1⁸⁹ᴳ* and *TAO3⁴⁴⁷⁷ᶜ* interact. A** Bar plots show the number of significantly expressed proteins compared with the S288c strain for the 0 h and 2 h 30 min. **B** GO enrichment analysis for upregulated genes in the MM and MMTT strains at 0 h and 2 h 30 min. Enrichment *p*-values were calculated using the cumulative hypergeometric test (one-sided) and adjusted for multiple comparisons using the Benjamini–Hochberg method (Metascape). Heatmaps show −log₁₀(*p*-value) for the top enriched GO terms. **C** Schematic representation of the arginine biosynthetic pathway. The proteins represented in blue are differentially upregulated during 2 h 30 min into sporulation. **D** Allocation of the whole cellular proteome to the arginine pathway in SS, MM, TT, and MMTT during the 0th hour and early stage of sporulation (2 h 30 min), calculated as the mean percentage allocation. Individual data points are shown as dots. The error bars represent the mean ± SD of three biological replicates. Created in BioRender. Sinha, H. (2025) https://BioRender.com/07eizi3.

regeneration through substrate-level phosphorylation even under conditions of limited mitochondrial activity. Consistently, Seahorse XF respirometry measurements revealed significantly higher basal oxygen consumption rates (OCR) in MM and MMTT strains compared to SS (Supplementary Fig. 13), aligning with the proteomic evidence of increased investment in mitochondrial respiration.

**Genetic interactions alter intracellular amino acid dynamics linked to nitrogen metabolism**

Through our transcriptomics and proteomics data, we revealed a distinct regulation of amino acid metabolism in the MMTT strain. Hence,

we wanted to test and show how the genetic interactions can reshape metabolic trajectories during sporulation. For this, we performed targeted temporal profiling of key intracellular amino acids across the SS, MM, TT, and MMTT strains at key developmental time points (0 h, 2 h 30 min, and 8 h) of sporulation (Fig. 6A).

Normalized intensity values were averaged across replicates and represented as z-score heatmaps. Clustering was performed on the MMTT strain to highlight its metabolic program, and the same amino acid order was retained across all strains for comparative analysis (Supplementary Fig. 14). We found that the MMTT strain showed a unique biphasic amino acid trend, which was not observed in the other

**Table 1 | Unique and shared upregulated proteins in MM, TT, and MMTT strains compared to the SS strain at 2 h 30 min into sporulation**

| Strains | Number of proteins | Differentially expressed proteins |
|---|---|---|
| MM, MMTT, TT | 1 | Syt1 |
| MM, MMTT | 24 | Arg3, Aro10, Cox1, Cox2, Cox5a, Cul3, Fmp10, Gre1, Gtt2, Ino1, Mef1, Mrp4, Mrpl17, Mrpl7, Mrps12, Mrps35, Msf1, Pbp1, Pbp4, Rec8, Rmd9, Ssa3, Sws2, Ykl065w-A |
| MMTT, TT | 3 | Bud23, Jhd2, Ypl245w |
| MMTT | 58 | Arg1, Arg4, Arg56, Arg8, Bcs1, Cbp4, Cit3, Cmc1, Cmc2, Cox13, Cox7, Dia4, Fmp33, Gcy1, Glg1, Gpx1, Ifm1, Mam33, Mcp2, Mho1, Mhr1, Mpm1, Mrp1, Mrp7, Mrpl10, Mrpl13, Mrpl3, Mrpl35, Mrpl4, Mrpl9, Mrps16, Mrps18, Mrps28, Mrps5, Mrps8, Mrps9, Msc6, Msk1, Mss116, Pam18, Pdh1, Ppa2, Rcf1, Rim4, Rnp1, Rsm10, Rsm23, Rsm24, Rsm26, Rsm28, Rsm7, Sct1, Slm5, Trm12, Vta1, Ydr115w, Ygr021w, Yml6 |
| MM | 17 | Aim32, Bop2, Cox23, Ctt1, Cyt2, Guf1, Itt1, Mdm35, Nam9, Pir3, Pot1, Pst1, Rsm19, Rsm27, Uga3, Yhr202w, Yjl045w |
| TT | 9 | Ahc1, Dal7, Hmi1, Htd2, Hym1, Lin1, Mnl1, Rme1, Yjr124c |

**Table 2 | Difference in summed protein allocation between 2 h 30 min and 0th hour in SS, MM, TT, and MMTT for translation, ribosome, and amino acid metabolism GO categories**

| Strains | Difference in summed mass protein allocation ($t_{2\ h\ 30\ min} - t_{0\ h}$) | | | |
|---|---|---|---|---|
| | Translation (%) | Ribosome (%) | Amino acid metabolism (%) | Arginine metabolism (%) |
| SS | −1.02 | −1.14 | 1.68 | 0.11 |
| MM | −1.42 | −0.94 | 0.77 | 0.17 |
| TT | −1.71 | −1.35 | 1.66 | 0.13 |
| MMTT | −2.52 | −1.91 | 1.46 | 0.24 |

strains (Fig. 6B; Supplementary Fig. 14). Notably, there was an early surge in the levels of alanine, arginine, lysine, glutamine, and histidine between 0 h and 2 h 30 min, followed by a pronounced depletion phase from 2 h 30 min to 8 h (Fig. 6B). This trajectory was consistent with our transcriptomic data, where we found a unique upregulation of genes related to amino acid metabolism, particularly histidine metabolism, arginine biosynthesis during the early phase (Supplementary Data 1). This shows an early biosynthetic burst, possibly driven by a programmed metabolic activation, followed by rapid mobilization of these resources for nucleotide biosynthesis during commitment to meiosis and spore morphogenesis. In contrast, the SS strain maintained a continuous accumulation of most amino acids, suggesting a dysregulation of intracellular metabolite utilization for meiosis and sporulation (Fig. 6C).

Additionally, glutamic acid, an early intermediate derived from acetate via the TCA cycle, was found to show a distinct early spike exclusively in the MMTT strain, followed by sustained levels up to 8 h (Fig. 6C). This pattern was absent in the other strains. The early accumulation of glutamate in the MMTT strain was consistent with previous studies reporting a transient increase in glutamate during early sporulation, which facilitated ammonium ion removal, a known inhibitor of the sporulation process[56].

Together, these data showed that amino acid metabolism was not merely passive during sporulation but was actively rewired in a strain-specific manner. The distinct temporal control of amino acid pools, especially those linked to nitrogen metabolism, emerged as a critical determinant of sporulation trajectory and efficiency, with the MMTT strain showing a dynamic and resource-intensive strategy to support its developmental program. Raw mass spectrometry intensity data for all measured amino acids are provided in Supplementary Data 7.

### Genetic interaction promotes the arginine biosynthetic pathway essential for mitochondrial function

From our multi-omics data analysis, we have identified a distinctive regulation of the arginine biosynthetic process in the MMTT strain during the initial stages of sporulation. This discovery led us to hypothesize that this regulation could have a significant causal role in modulating sporulation efficiency, mainly through genetic interactions involving the $MKT1^{89G}$ and $TAO3^{4477C}$ SNPs. Previous studies on other yeast strains, such as SK1 and W303, known for their high sporulation efficiency, have shown an upregulation of the $ARG4$ gene during early sporulation phases[35]. However, $arg4\Delta$ in the S288c strain, which typically exhibited lower sporulation efficiency, did not appear to affect the sporulation process. This phenotypic divergence among strains prompted us to investigate whether a unique interaction between the $MKT1^{89G}$ and $TAO3^{4477C}$ SNPs in the MMTT strain could modulate sporulation through the arginine biosynthesis pathway. To test this, deletions of $ARG4$, argininosuccinate lyase, a key enzyme involved in the final step of the arginine biosynthetic pathway, and $ARG56$, acetyl glutamate kinase and N-acetyl-gamma-glutamyl-phosphate reductase, catalyzing the second and third steps in the arginine biosynthetic pathway, in SS, MM, TT, and MMTT strains were generated. Our working hypothesis was that if the arginine biosynthetic pathway served as a causal route responsible for the observed additive phenotypic effects of the $MKT1^{89G}$ and $TAO3^{4477C}$ SNPs on sporulation efficiency, then the deletion of these key genes would have a more pronounced impact on the MMTT strain than on the other strains.

Sporulation efficiency of wild-type and deletion strains showed that $arg4\Delta$ and $arg56\Delta$ did not affect sporulation efficiency in SS, MM, and TT strains. In contrast, these deletions showed no sporulation phenotype, specifically in the MMTT strain with $arg4\Delta$, even after 48 h in the sporulation medium (Fig. 7A). This strain-specific phenotype suggested a potential link between $arg4\Delta$ and a defect in mitochondrial stability or respiration, crucial for the initiation of sporulation.

Since respiration is closely linked to the sporulation process, especially under nutrient-limiting conditions, we aimed to investigate the underlying respiratory dysfunction in the MT strain further. Growth phenotype of S, M, T, and MT strains and their $arg4\Delta$ in non-fermentable carbon sources, such as glycerol, ethanol, and glycerol/ethanol mixtures (YPG, YPE, and YPEG) showed that the S, M, and T strains showed no growth defects in these media. However, despite an extended incubation period of four days, MT-$arg4\Delta$ failed to grow in these non-fermentable carbon sources (Fig. 7B). This indicated that $arg4\Delta$ in the MT strain showed a strong respiratory defect. Further, this growth and sporulation defect in MMTT-$arg4\Delta$ was not rescued even with supplementing arginine and other amino acids (Supplementary Fig. 15).

To find if the observed respiratory defect was linked to impaired mitochondrial activity, we performed a Mitotracker assay to measure mitochondrial activity in these strains. The assay was performed for 2 h after transferring haploid cells to a non-fermentable carbon source (acetate). The Mitotracker assay results showed that the MT strain with $arg4\Delta$ significantly reduced mitochondrial activity compared to its wildtype ($p$-value < 0.001), indicating that $ARG4$ was required to

**A**                **Targeted intracellular amino acid data generation**

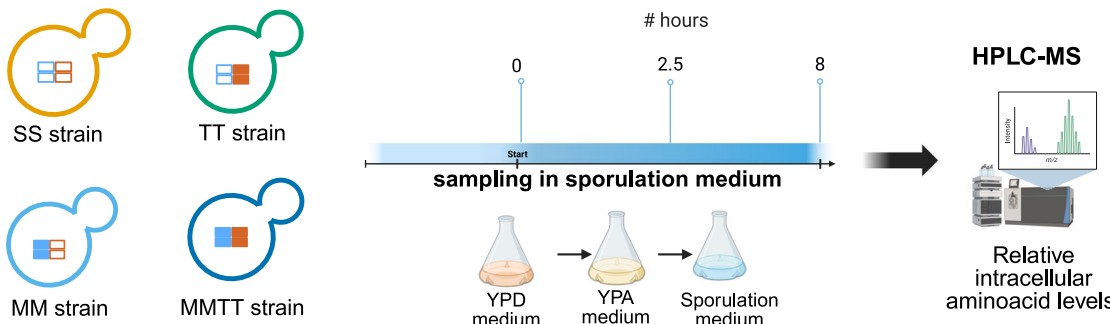

**B** Amino acid levels that show accumulation followed by depletion in MMTT strain
(Cluster 1, 2 and 8)

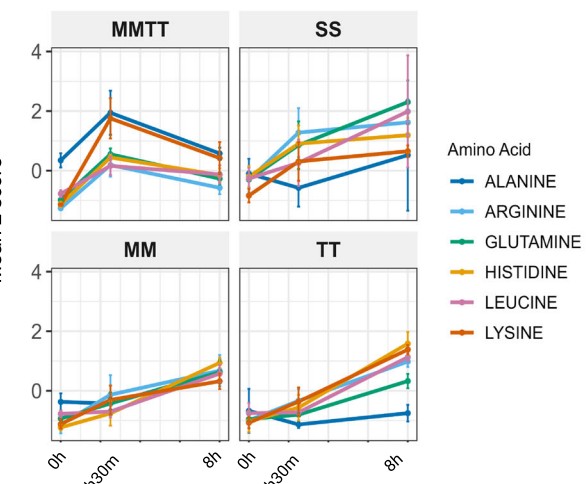

**C** Amino acid levels that show accumulation followed by steady profiles in MMTT strain
(Cluster 3 and 6)

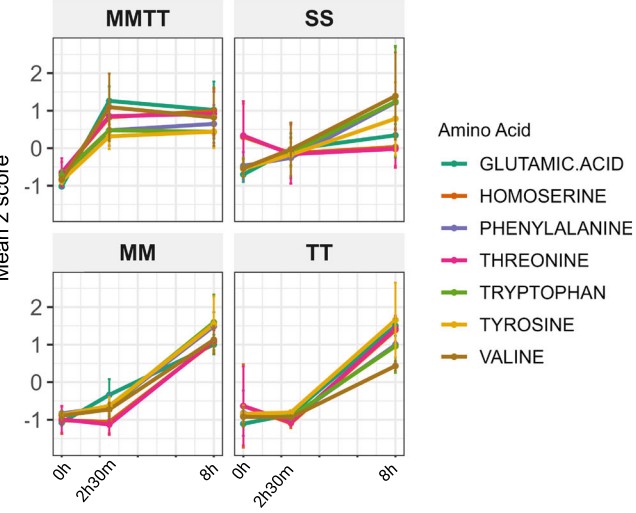

**Fig. 6 | Genetic interactions alter intracellular amino acid dynamics linked to nitrogen metabolism. A** The SS, MM, TT, and MMTT yeast strains were grown in sporulation medium, and samples were collected at 0 h, 2 h 30 min, and 8 h during sporulation. Intracellular amino acid levels were analyzed using mass spectrometry. **B** z-score profiles of selected amino acids in clusters 1, 2, and 8 (alanine, histidine, glutamine, leucine, arginine, and lysine) at 0 h, 2 h 30 min, and 8 h across SS, MM, TT, and MMTT strains. **C** z-score profiles of selected amino acids in clusters 3 and 6 (tyrosine, tryptophan, phenylalanine, asparagine, valine, glutamic acid, threonine, homoserine) at 0 h, 2 h 30 min, and 8 h across SS, MM, TT, and MMTT strains. The clusters are given in Supplementary Fig. 14. The error bar represents mean ± SD of three biological replicates (**B**, **C**). Source data is provided as a Source Data file for (**B**, **C**). Created in BioRender. Sinha, H. (2025) https://BioRender.com/07eizi3.

maintain mitochondrial function in the MT strain. In contrast, as expected, the S, M, and T strains did not show a reduction in mitochondrial activity following *arg4Δ* (Supplementary Fig. 16). We also measured mitochondrial activity in SS, MM, TT, and MMTT diploid strains after 2 h of incubation in the sporulation medium. Consistent with the results from haploid cells, we found that only the MMTT strain with *arg4Δ* exhibited a pronounced reduction in Mitotracker intensity compared to the wild-type MMTT strain (Fig. 7C). This suggested the essential role of *ARG4* in maintaining mitochondrial function, specifically in the MMTT strain. We also found that *arg4Δ* in the MMTT strain resulted in only petite colonies, which suggested that *ARG4* was required to maintain mitochondrial stability (Fig. 7D).

We then measured basal respiration rates in fermentable and nonfermentable media to further probe the effects of *arg4Δ* and *arg56Δ* on respiratory function. For this, we assessed the oxygen consumption rates (OCR) using a Seahorse assay of all strains in acetate and glucose media, as acetate is known to be a respiratory substrate. The results revealed that neither *arg4Δ* nor *arg56Δ* significantly impacted OCR in the SS and TT strains, indicating that respiration in these strains remains unaffected mainly by disruptions in the arginine biosynthetic pathway (Fig. 7E). However, in the MM strain, both deletions led to a reduction in OCR, though this reduction did not impair sporulation efficiency, suggesting that compensatory mechanisms support sporulation independently of respiration. Only in the MMTT strain did *arg4Δ* completely abolish respiratory function, while *arg56Δ* had no significant effect on respiration (Fig. 7E). Interestingly, these patterns were not observed when glucose was used as the carbon source, highlighting the respiratory-specific impact of *arg4Δ* (Supplementary Fig. 17).

In conclusion, our findings demonstrated that the arginine biosynthesis pathway was critical in initiating sporulation and maintaining mitochondrial function only in the MMTT strain.

## Discussion

Genome-wide gene deletion studies have shown that the combination of gene deletions can be additive, and epistatic interactions have

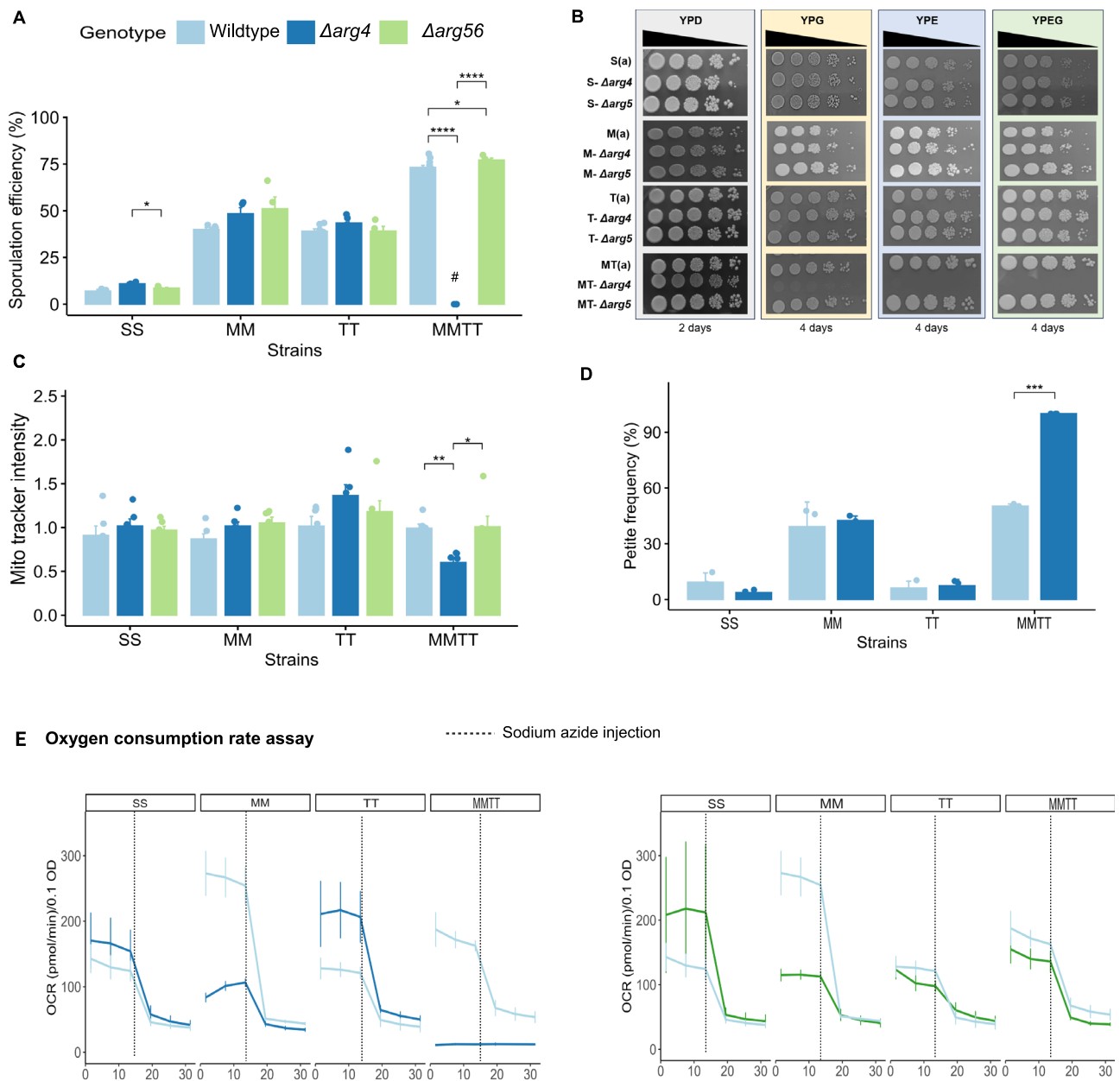

**Fig. 7 | The arginine biosynthetic pathway is essential to maintain mitochondrial function when *MKT1*[89G] and *TAO3*[4477C] interact. A** Sporulation efficiency was calculated after 48 h for wildtype (SS, MM, TT, and MMTT) and their respective *arg4Δ and arg56Δ*. # represents no sporulation (*N* = 3). **B** 10-fold serial dilution of wildtype, *arg4Δ, arg56Δ* haploid strains of S288c, M, T, and MT strains were spotted on YPD (2%), YPE (2%), YPG (2%), and YPEG. The pictures were taken after two days for YPD and four days for YPE, YPG, and YPEG hours of incubation at 30 °C. **C** Mitotracker intensity of diploid strains of SS, MM, TT, and MMTT (wild-type and *arg4Δ*) after 2 h incubation in sporulation medium (1% potassium acetate) (*N* = 4). **D** Petite frequency assay of wild-type and *arg4Δ* of the MMTT strain (*N* = 3). **E** The oxygen consumption rate (OCR) of SS, MM, TT, and MMTT for WT (*N* = 4), *arg4Δ* (*N* = 4), and *arg5Δ* (*N* = 3) cells was measured using the Seahorse Extracellular Flux 96 Analyzer, grown in acetate media. After three basal measurements, sodium azide was injected into the wells to shut off mitochondrial oxygen consumption, and an additional three sets of measurements were taken. *p*-values were calculated using a two-sided unpaired *t*-test in the rstatix R package (v.0.7.2) (**A**, **C**, **D**). Significance levels are indicated as follows: ****$p < 0.0001$, ***$p < 0.001$, **$p < 0.01$, *$p < 0.05$. The exact *p*-values are provided in the source data file (**A**, **D**). The error bar represents the mean ± SD (**A**, **C–E**). Source data are provided as a Source Data file for (**A**, **C–E**). Created in BioRender. Sinha, H. (2025) https://BioRender.com/07eizi3.

different phenotypic effects than the effect of individual gene deletions[6,63,64]. We extend this concept to study natural genetic variants and show that when two additive effect SNPs are present together in a strain, they impact the phenotype by activating a unique metabolic pathway different from the metabolic pathways of individual SNP effects. Our result has consequences for understanding how variants affect disease phenotypes, specifically in complex traits where multiple SNP combinations can have differential outcomes in the progression of the disease and treatment responses.

Complex trait variation arises from both genetic and environmental influences. When multiple causal genes contribute to a trait, the resulting phenotype can be shaped by gene–gene–environment (G×G×E) interactions. To isolate the role of gene–gene (G×G) interactions, we held the environment constant and examined the consequences of genetic interactions to the phenotypic variation by studying the intermediate phenotypes like gene expression, protein expression, and metabolite levels. Using a multi-omics approach, combining temporal transcriptome, proteome, and metabolite

profiling of isogenic allele replacement yeast strains, we were able to ascertain that the strain with combined SNPs, $MKT1^{89G}$ and $TAO3^{4477C}$, activated the amino acid metabolism pathways early in sporulation while simultaneously downregulating ribosome-related processes. This observation suggested a metabolic trade-off wherein yeast reduces energy-intensive processes like ribosome biogenesis and redirects resources toward amino acid biosynthesis in response to nutrient starvation[65]. Further analysis of proteome allocation, supported by intracellular ATP and respiratory assays, revealed substantial reprogramming of mitochondrial function and biogenesis during early adaptation to respiratory metabolism. These findings are consistent with previous reports by Björkeroth et al.[53]. We also found that the strains with both SNPs in combination can activate the arginine biosynthetic pathway differentially by allocating more protein to arginine biosynthesis than to ribosome biogenesis and glycolysis. While we acknowledge that a complete energy budget for protein turnover was not captured in our dataset, the absolute proteomic quantification provided a critical and biologically meaningful readout of resource allocation trends. These trends are especially relevant in developmental programs like sporulation, where resource prioritization, rather than growth optimization, becomes the dominant strategy. Thus, our protein allocation analysis provided a first-order approximation of how proteome resources were being reallocated during critical transitions, such as sporulation. Further, through experimental validation, we have shown that $ARG4$ is necessary to maintain mitochondrial activity and respiration only when $MKT1^{89G}$ and $TAO3^{4477C}$ SNPs are combined. We speculate that due to the genetic interactions between $MKT1^{89G}$ and $TAO3^{4477C}$ SNPs, a rewiring of the metabolic network makes mitochondrial function heavily dependent on the arginine biosynthetic pathway. This was further supported by the unique biphasic dynamics of alanine and arginine in the MMTT strain, which accumulated early (0 h–2 h 30 min) and was rapidly depleted by 8 h, unlike in other strains. The increased sporulation efficiency in the MMTT strain could be attributed to the immediate response to nutrient starvation and their ability to synthesize the necessary amino acids like histidine during the early phase as precursors for nucleotide biosynthesis and meiotic process during the later stages. The interaction observed here shares conceptual parallels with known "moonlighting" functions of metabolic enzymes. For example, Alt1, an alanine transaminase, has been implicated in mitochondrial gene regulation beyond its catalytic activity[66]. Similarly, Ilv5, involved in branched-chain amino acid biosynthesis, also contributes to mtDNA stability[67]. These findings indicate that metabolic enzymes, including those in the arginine pathway, might participate in mitochondrial regulation beyond their canonical roles, particularly under conditions of network rewiring imposed by genetic variation[68].

The observed allele frequencies of $MKT1^{89G}$ and $TAO3^{4477C}$ indicate distinct evolutionary dynamics. The near-fixation of $MKT1^{89G}$ in *S. cerevisiae* suggests it confers an adaptive advantage, particularly in stress-related contexts, as shown in prior studies[69,70]. In contrast, the $TAO3^{4477C}$ variant is extremely rare, implying it may be deleterious in most backgrounds or advantageous only under specific environmental or genetic conditions while having an effect size similar to a common variant[71]. Notably, when $TAO3^{4477C}$ co-occurs with $MKT1^{89G}$, it induces a metabolic shift activating arginine biosynthesis and suppressing ribosome biogenesis, indicating a context-dependent benefit that arises from this genetic interaction.

Beyond yeast, these findings provide a conceptual framework for understanding how genetic interactions (G×G) can reconfigure cellular metabolism to activate latent pathways. In human systems, genetic interactions further modified by environmental factors may activate latent pathways that can influence disease phenotypes or therapeutic responses, especially in conditions where metabolic reconfiguration and rare allele effects are observed. Our work emphasizes the importance of resolving variant effects in combinatorial contexts and

demonstrates how integrated multi-omics approaches can uncover the molecular logic of complex trait architecture.

## Methods

### Allele replacement strains

All yeast strains are derivatives of S288c, a widely used laboratory strain, and their relevant genotypes are listed in Supplementary Table 1. The available diploid MM strain with $MKT1^{89G}$ allele and TT strain with $TAO3^{4477C}$ allele, corrected for background mutations, were validated for their SNPs using allele-specific PCRs and Sanger sequencing. The haploids of M and T strains were generated by sporulating their respective diploid strains, followed by spore enrichment. The resulting M and T haploids were confirmed using the MAT locus PCR. The opposite mating types of M and T strains were crossed to obtain the diploid MmTt strain with genotype $MKT1^{89G}/MKT1^{89A}$ $TAO3^{4477C}/TAO3^{4477G}$. The diploid MmTt strain was sporulated and spore-enriched, which resulted in segregants having all possible combinations of $MKT1$ and $TAO3$ alleles: Mt ($MKT1^{89G}/TAO3^{4477G}$), mt ($MKT1^{89A}/TAO3^{4477G}$), mT ($MKT1^{89A}/TAO3^{4477C}$), and MT ($MKT1^{89G}/TAO3^{4477C}$). The MT haploids were selected by performing allele-specific PCRs and were further validated using Sanger sequencing. The opposite mating types of MT haploids were mated to obtain the diploid MMTT strain with genotype $MKT1^{89G}/MKT1^{89G}$ $TAO3^{4477C}/TAO3^{4477C}$. All further experiments are performed using the diploid MMTT strain.

### Phenotyping for sporulation efficiency

The diploid SS, MM, TT, MmTt, MMTT, and SK1 strains were phenotyped for sporulation efficiency. Briefly, the strains were first grown in YPD (2% (w/v) yeast extract (HIMEDIA, Cat. No. CR027), 1% (w/v) peptone (HIMEDIA, Cat. No. CR001), 2% (w/v) dextrose (HIMEDIA, Cat. No. PCT0603)) from $OD_{600}$ 0.2 to 1.0 and then grown in YPA (1% (w/v) yeast extract, 2% (w/v) peptone, 1% (w/v) potassium acetate (Sisco Research Laboratories, Cat. No. 96248)) from $OD_{600}$ 0.2 to 1.0. The cells were washed in water, then in potassium acetate, and then incubated in 1% potassium acetate supplemented with amino acids (supplemented with 20 μg ml$^{-1}$ uracil (Sigma, Cat. No. U0750), 20 μg ml$^{-1}$ histidine (Sigma, Cat. No. H8000), 30 μg ml$^{-1}$ leucine (Sigma, Cat. No. L8000), 20 μg ml$^{-1}$ methionine (Sigma, Cat. No. M9625) and 30 μg ml$^{-1}$ lysine (Sigma, Cat. No. L5626)). The sporulation efficiency was calculated as the ratio of dyads and tetrads produced by a strain to the number of single-nucleus cells.

### Generation of *ARG4 and ARG56* deletion mutants

The gene deletions of *ARG4* and *ARG56* were performed in haploids of S, M, T, and MT by replacing the *ARG4* and *ARG56* genes with drug resistance cassettes (*hph*MX4) by following the high-efficiency lithium acetate protocol[72]. The transformants were selected by plating onto YPD media containing hygromycin (HIMEDIA, Cat. No. A015). The deletion confirmation (homologous integration) was done by colony PCR of insertion junctions using junction-specific primers. The deletion confirmed haploid cells were diploidized by transforming strains with the pHS03 plasmid (resistant to Nat (JENA Bioscience, Cat. No. AB-102)) and confirmed using the MAT locus PCR. The list of primers used in this study is given in Supplementary Table 2. Further, the diploid strains with respective deletions were phenotyped for sporulation efficiency and for studying mitochondrial activity.

### Spot dilution assay

Overnight cultures of the strains were grown in 5 ml of YPD and diluted to an $OD_{600}$ of 1.0. Ten-fold serial dilutions were prepared, and 5 μl of each dilution was spotted onto fermentable media (YPD) and non-fermentable carbon source media (YPGlycerol (2%), YPEthanol (2%), YPEG (1% ethanol (Hayman, Cat. No. F204325) + 3% glycerol (HIMEDIA, Cat. No. TC503))). The final $OD_{600}$ values of the spots were 1, 0.1, 0.01,

0.001, and 0.0001. Plates were incubated at 30 °C, with growth captured after 2 days for YPD and 4 days for the non-fermentable media.

## Mitotracker fluorescence assay

The Mitotracker plate reader assay was adapted from Vengayil et al.[73] with modifications for high-throughput analysis. In brief, overnight yeast cultures were initially grown in YPD medium, followed by secondary cultures diluted to an $OD_{600}$ of 0.2 and grown to an $OD_{600}$ of 1.0. Cells were washed and incubated in 1% potassium acetate for 2 h. Following this, Mitotracker CMXRos (ThermoFisher, Cat. No. M7512) was added to each well to a final concentration of 200 nM, and the plates were incubated in the dark for 30 min at 30 °C in a shaking incubator. After incubation, the plates were centrifuged at 1700 rpm for 2 min, and the supernatant was discarded. The cells were washed once with PBS (pH 7.4, HIMEDIA, Cat. No. TS1006) to remove excess dye and media, and 200 μl of PBS was added to each well. The cells were resuspended and transferred to black plates for fluorescence measurement. Fluorescence was recorded using a multimode plate reader (BioTek Synergy H1, Agilent Technologies Inc.) at excitation/emission wavelengths of 572/599 nm. Finally, the fluorescence signal was normalized using the optical density ($OD_{600}$) of the same samples, which was measured.

## Petite frequency assay

Three independent colonies were resuspended in 1 ml of PBS, diluted, and plated on YPDG medium (1% yeast extract, 2% peptone, 0.1% glucose, and 3% glycerol) to yield approximately 200–400 colonies per plate. After five days of incubation, colonies were counted, distinguishing between large colonies (grande) and small colonies (petite). The petite frequency was calculated as the ratio of petite colonies to the total number of colonies per plate. Petite colonies were further validated by patching them onto YPE plates to confirm their inability to grow on this medium.

## Oxygen consumption rate assay

Yeast oxygen consumption rate (OCR) measurements were performed using the Agilent Seahorse XF Pro analyzer, following a protocol specifically optimized for 0.5% sodium azide (Sigma, Cat. No. S-2002) injection, as outlined in Yao et al.[74] and Walden et al.[75] On the first day, overnight yeast cultures (consisting of 4 biological replicates each for SS, MM, TT, and MMTT and four biological replicates for *arg4* deletion strains and three biological replicates for *arg5* deletion strains) were prepared. The Seahorse XF96 sensor cartridge was pre-incubated with 200 μl of calibrant solution per well at 30 °C in a CO$_2$-free incubator. The Seahorse assay media containing glucose and acetate (0.167% yeast nitrogen base, 0.5% ammonium sulfate, and 1% acetate or 2% dextrose, respectively) was prepared. The microplate wells were coated with 60 μl of poly-L-lysine solution (Sigma-Aldrich, Cat. No. P4707) (0.1 mg ml$^{-1}$), incubated for 1 h, washed twice with PBS, and stored at room temperature overnight. On the second day, overnight yeast cultures grown in YPD were diluted and grown until reaching an $OD_{600}$ of 0.5–0.7 (mid-log phase). The cells were washed and diluted to a specific concentration in Seahorse assay media (glucose or acetate). The cells were seeded into poly-L-lysine-treated wells of the XF96 microplate, then centrifuged at 500 rpm for 5 min and incubated at 30 °C for 1 h. The sensor cartridge was loaded with 0.5% sodium azide in chamber A and was calibrated. Three basal OCR measurements were recorded at 30 °C for 3 min each, with 2 min of mixing between readings. Following the injection of 0.5% sodium azide into chamber A, three additional OCR measurements were taken, following the same procedure. The measured OCR values were then normalized to $OD_{600}$ 0.1.

## Extracellular acetate analysis using HPLC

Two biological replicates of SS, MM, TT, and MMTT strains were grown in YPD, followed by growth in YPA and then transferred to 1%

potassium acetate. Samples $OD_{600}$ of 1.0 in sporulation medium were collected at appropriate time points, and the extracellular medium was filtered using 0.22 μm filters and stored at −20 °C until analysis. Extracellular acetate levels in the sporulation medium were measured using Shimadzu P-series Quaternary Gradient High Performance Liquid Chromatography equipped with a Phenomenex Rezex ROA-Organic acid H+ (8%) column [300 × 7.8 mm] and an RI detector. The elution buffer used was 5 mM H$_2$SO$_4$ (Finar, Cat. No. 7664-93-9), with a flow rate of 0.6 ml min$^{-1}$, and the column was maintained at an oven temperature of 40 °C.

## Intracellular acetate extraction and quantification

Yeast cells equivalent to an $OD_{600}$ of 1.0 were collected at defined time points from the sporulation medium. Pellets were washed twice with ice-cold PBS, snap-frozen in liquid nitrogen, and stored at −80 °C until extraction. For acetate extraction, frozen pellets were thawed and resuspended in 250 μl of ice-cold 0.1 N HCl (Sisco Research Laboratories, Cat. No. 34472). Cells were lysed by vortexing for 5 min with the addition of acid-washed glass beads (Sigma, Cat. No. G8772). The lysates were centrifuged at 14,000 rpm for 10 min at 4 °C, and 180–250 μl of the clear supernatant was transferred to a fresh microcentrifuge tube. The pH of the extract was adjusted to 6.5–7.0 using 3–5 μl of 5 N NaOH (Merck, Cat. No. 193102) with gentle mixing. The neutralized cell extracts were used for acetate quantification following the manufacturer's instructions provided with the Megazyme Acetic Acid Rapid Kit (Megazyme, Ireland, Cat. No. K-ACETRM). Measurements were performed using freshly prepared extracts, with three biological replicates per condition.

## Quantification of intracellular ATP levels

Intracellular ATP levels were quantified in yeast strains SS, MM, TT, and MMTT, using a minimum of two biological replicates per strain. Samples were collected at four time points (0 h, 1 h 10 min, 2 h 30 min, and 8 h) following transfer to sporulation medium at an $OD_{600}$ of 1.0. Cells were rapidly quenched in −80 °C pre-chilled 100% methanol and pelleted by centrifugation at 10,000 × g for 1 min at 4 °C. The pellets were washed once with −80 °C methanol (HIMEDIA, Cat. No. MB113) and stored at −80 °C until further processing.

ATP was extracted using an acetone-based method as previously described in Takaine et al.[76], with minor modifications. Briefly, cell pellets were resuspended in 0.75 ml of 90% acetone (Sisco Research Laboratories, Cat. No. 31566) and mixed by repeated pipetting and brief vortexing. Samples were then placed in a fume hood on a dry block heater with open lids and incubated at 90 °C for 15 min to allow complete evaporation of acetone. Following evaporation, samples were centrifuged at high speed for 15 s. The resulting aqueous supernatant (≤40 μl) was transferred to a new microcentrifuge tube and further clarified by centrifugation at high speed for 3 min at 4 °C.

ATP concentrations were measured using a luminescence-based detection assay (ATP Assay Kit, Sigma-Aldrich, Cat. No. MAK473) according to the manufacturer's instructions. Luminescence was recorded using white opaque flat-bottom 96-well plates on a multimode microplate reader (BioTek Synergy H1, Agilent Technologies Inc.). ATP concentrations were determined using a standard curve generated with known ATP standards provided in the kit.

## Temporal transcriptome profiling and data pre-processing

The SS and MMTT strains were first grown in YPD and then in YPA and transferred to the sporulation medium (1% potassium acetate and amino acid supplements). The samples were taken at 0 h, 30 min, 45 min, 1 h 10 min, 1 h 40 min, 2 h 30 min, 3 h 50 min, 5 h 40 min, and 8 h 30 min and were snap-frozen and stored at −80 °C. RNA extraction, quality control, library preparation, and paired-end sequencing using Illumina Novaseq were conducted by an external service provider (Genotypic Technology Pvt Ltd, Bangalore, India). The detailed

protocols followed for RNA isolation and library preparation are given in the Supplementary Methods section.

The raw Fastq files were given as input to a customized Snakemake pipeline, which performs quality checks using FASTQC and MultiQC[77], removes rRNA contamination, performs alignment using STAR[78], and finally gives the feature counts as output. The feature count file was then provided as input to the IDEP 2.0 software[79] for initial quality checks. The low counts were filtered such that three counts per million (edge R) were in at least three samples. Thus, 6207 genes from an initial list of 7127 genes were selected for further analysis. Using the 300 most variable genes, hierarchical clustering and PCA were performed for all 54 samples, which removed five potential outlier samples from further analysis.

### Differential expression analysis using DESeq2 in IDEP software

Initially, we aimed to independently identify genes exhibiting significant differential expression relative to the baseline (0 h) at each time point for both MMTT and SS strains. Our dataset comprised 24 samples for the MMTT strain, encompassing 7127 genes. Subsequently, we filtered this dataset to retain 6447 genes with a minimum count per million (CPM) threshold of 0.5 across at least three samples. Employing the DESeq2[51] algorithm with a false discovery rate (FDR) cutoff of 0.05 and a minimum fold change of 2, along with the Wald test and independent filtering, we identified genes showing significant differential expression. Similarly, in the S strain, we analyzed a dataset containing 6409 genes that passed the aforementioned filtering criteria. We used DESeq2 under identical parameters to identify genes exhibiting differential expression relative to the 0th hour time point.

### Temporal differential expression analysis using the likelihood ratio test

We further investigated temporal differential gene expression employing the Likelihood Ratio Test (LRT) method within DESeq2. This approach evaluates whether significant differences in genotype effects exist across various time points. To construct the 'full model,' we integrated key sources of variation, including genotype and time, along with our condition of interest, i.e., the interaction between genotype and time (genotype × time). By contrasting this full model with a 'reduced model' lacking the 'genotype x time' term, we identified genes exhibiting significant alterations in expression profiles between SS and MMTT strains across multiple time points. This analysis enabled us to pinpoint genes whose expression trajectories were influenced by the presence of $MKT1^{89G}$ and $TAO3^{4477C}$.

### Temporal gene expression clustering using DPGP

We utilized gene expression data normalized to Transcripts Per Million (TPM) and provided it as input to the DPGP algorithm[52]. To optimize computational efficiency, we employed the '--fast' option, which utilizes an accelerated computation mode. Moreover, we utilized the '--true_times' parameter to preserve the temporal information associated with each expression profile during the clustering process due to non-uniform sampling time points in our data. This enabled us to capture the temporal dynamics inherent in the data accurately.

### Absolute quantitative proteomics using mass spectrometry

To capture the absolute proteome changes during the initial phases of sporulation, where $MKT1^{89G}$ and $TAO3^{4477C}$ SNPs were found to be active, we performed absolute proteomics experiments for all four strains at two time points. For this, 4 ml of the diploid strains SS, MM, TT, and MMTT at $OD_{600}$ of 1.0 were sampled at 0 h and 2 h 30 min after incubation in sporulation medium. The samples were centrifuged at 4 °C for 2 min at $4000 \times g$. The supernatant was discarded, and the cell pellets were snap-frozen in dry ice and stored at −60 °C until further analysis. Sample preparation for proteomics analysis was performed as described previously by Kozaeva et al.[80], with some modifications (see Supplementary Methods for details)[81].

### Processing of mass spectrometric raw data

Sequence identification was performed for all analyses using a protein database consisting of the *S. cerevisiae* S288c (UP000002311; accessed on the 17th of November 2023) reference proteome. Spectronaut V18 was used for protein identification and relative quantification of peptides for DIA data analysis. The default settings for "directDIA" were applied with an FDR cutoff of 1%, except for MS1 quantification for the peptides. Protein abundances were inferred from the peptide abundances using the xTop algorithm[82]. Absolute protein quantification was performed through the TPA (total protein approach)[83] method to the final units of fmol $\mu g^{-1}$ total protein as proteome composition values.

### Differential protein expression and allocation analysis

We converted the fmol $\mu g^{-1}$ total protein to the percentage of the entire protein mass (g $\mu g^{-1}$ total protein). This was done by multiplying by molecular weight and dividing by the sum of all proteins. This analysis expressed every protein abundance as a percentage of the entire proteomic mass. These mass percentages were then used for subsequent protein allocation and differential abundance analysis. For GO-slim mapper process terms analysis, all proteins in all datasets were merged, and the terms that matched the proteins were annotated to them as described in Björkeroth et al.[53]. For the summation of allocation, each dataset was subsequently matched to the set-up GO-slim mapper process term framework to identify which proteins are to be summed. The differential expression of proteins between wildtype (SS) versus allele replacement strains (MM, TT and MMTT) and also between the time points (0 h versus 2 h 30 min) was performed using the $\log_2$-transformed values with unpaired two-sided Student's *t*-test. The significantly reallocated proteins are proteins with a $\log_2$(fold change) greater than +1 or less than −1 and a *p*-value < 0.05 for all pairwise comparisons performed using the dataset.

### mRNA-protein correlation analysis

To understand how mRNA correlates with the protein levels, we computed the Pearson correlation, plotting the average $\log_2$(TPM) for mRNA against the average $\log_2$(abundance) protein. The cor.test() function from the R 'stats' package was employed to assess the statistical significance of correlations. Further, the mRNA-protein slope was identified from the coefficient derived from the linear model (lm() function in R's 'stats' package) using the formula 'lm(protein ~ mRNA)', where 'protein' is the $\log_2$(abundance) and 'mRNA' is the $\log_2$(TPM).

### LC-MS analysis for relative intracellular amino acid quantification

Samples were collected at three time points (0 h, 2 h 30 min, and 8 h) following transfer to sporulation medium at an $OD_{600}$ of 1.0. Metabolism was rapidly quenched by adding four volumes of ice-cold 60% methanol (kept in −80 °C), followed by incubation on ice for 5 min. Cells were pelleted by centrifugation at $1000 \times g$ for 3 min at 0 °C and washed once with 700 $\mu l$ of ice-cold 60% methanol (kept in −80 °C). The pellet was then resuspended in 1 ml of 75% ethanol and incubated at 80 °C for 3 min to extract intracellular metabolites. Samples were cooled on ice for 5 min and centrifuged at maximum speed for 10 min at 4 °C. Supernatants were collected, dried in a vacuum concentrator, and stored at −80 °C until analysis. LC-MS/MS analysis was performed on an Agilent 6495 Triple Quadrupole mass spectrometer operating in positive ionization mode using multiple reaction monitoring (MRM) by C-CAMP Bengaluru, India. The sample processing and detailed protocol are given in the Supplementary Methods.

### Generation and analysis of context-specific models

We employed a genome-scale metabolic modeling approach to understand how the SNPs and their interactions can influence the intracellular flux variation during the early sporulation phase. We generated context-specific metabolic models by integrating the protein expression data with the yeast genome-scale metabolic model (Yeast9)[84]. Further, we analyzed the intracellular flux variation between the generated models using genome-scale differential flux analysis (GS-DFA) as described in the Supplementary Methods section (Supplementary Note 1, Supplementary Data 8).

### Reporting summary

Further information on research design is available in the Nature Portfolio Reporting Summary linked to this article.

## Data availability

Fastq files and raw gene-count data are available on GEO under the accession number GSE278267. The mass spectrometry proteomics data have been deposited in the ProteomeXchange Consortium via the PRIDE[85] partner repository with the dataset identifier PXD056947. The metabolomic MS raw data have been deposited in MetaboLights with the dataset identifier MTBLS12761. All other data supporting the findings of this study have been provided as supplementary tables and source data files. Source data are provided with this paper.

## Code availability

All analyses were performed using R version 4.3.1. All figures were generated using custom R code. The R codes used for analyses are available on GitHub [https://github.com/HimanshuLab/molecular-additivity-of-QTNs] under MIT license and archived at Zenodo [https://doi.org/10.5281/zenodo.13917859][86].

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

## Acknowledgements

We thank Veerendra Gadekar (IIT Madras) and Venkatesh K (IIT Madras) for their assistance with computational analysis and for valuable discussions. We also appreciate the insightful discussions and support from all members of the Systems Genetics Lab, Centre for Integrative Biology and Systems Medicine (IBSE), and Wadhwani School of Data Science and Artificial Intelligence (WSAI). We acknowledge the Sustainable Bioprocessing Laboratory, IIT Madras, for the HPLC facility. We acknowledge Divya Dharshini (Technical University of Denmark) and Avinash Saravanan (IIT Madras) for their help with HPLC. Special thanks to Varsha Goyal (IIT Madras) for her contributions to the experiments during the revision process. We acknowledge MS Facility, C-CAMP, Bengaluru, for their service related to amino acid quantification and the BIO-SAIF facility, Department of Biotechnology, IIT Madras, for the multimode microplate reader (BioTek Synergy H1) and Agilent Seahorse XF Analyzer (Agilent Technologies Inc.). Computational analysis was carried out using resources provided by IBSE, IIT Madras. Additionally, S. Sasikumar acknowledges support through the HTRA fellowship from IIT Madras and Excelra Knowledge Solutions Private Limited. S. Sasikumar also acknowledges a travel fellowship from IBSE, IIT Madras. We acknowledge core funding to IBSE, IIT Madras (BIO/1819/304/ALUM/KARH), the Centre of Excellence funding to IBSE (SB/2021/0841/BT/MHRD/08752), and Excelra Knowledge Solutions Private Limited (CR/22-23/0026/BT/EXCE/008752) to H.S. For cultivation, HPLC, and proteome analysis, we acknowledge the funding received from The Novo Nordisk Foundation (NNF20CC0035580) to S.S.

## Author contributions

H.S. conceptualized the study. S. Sasikumar generated strains, performed the experiments, analyzed multi-omics data and results, developed GSMM modeling, prepared illustrations, and wrote the first draft. S. Sasikumar and H.S. wrote the manuscript. S.T.P. performed proteomics experiments and analyzed mass spectrometry data. H.S. and S. Sudarsan supervised and acquired the funding. All authors contributed to data interpretation and provided feedback on the manuscript. All authors approved the final version of the manuscript.

## Competing interests

The authors declare no competing interests.
