## [Transparent Peer Review file · Nature Communications]

Interaction of Genetic Variants Activates Latent Metabolic Pathways in Yeast

Corresponding Author: Dr Himanshu Sinha

Version 0:

Reviewer comments:

Reviewer #1

(Remarks to the Author)

Genetic interactions, essential in shaping complex traits across species, are often poorly understood. This study investigates how interactions between two SNPs, MKT189G and TAO34477C, in *Saccharomyces cerevisiae* influence sporulation. By integrating gene expression and proteomics data, the authors reveal that these SNPs activate the arginine biosynthesis pathway while suppressing ribosome biogenesis under nutrient starvation, highlighting a metabolic trade-off. This SNP interaction enhances sporulation by modulating cellular stress responses. The study also connects these findings to human diseases like cancer, where similar SNP combinations may disrupt metabolic pathways, leading to uncontrolled cell proliferation.

This is interesting work leading to a more precise view of key aspects of genetic interaction at different levels. The question is clearly posed and the analyzes give interesting results. This study is important because it provides strong experimental data to support the understanding of one of the keys to the genotype-phenotype relationship. This is a well-conducted study.

Overall, this is very nice story with interesting conclusions.

I just have a couple of minor points that need to be addressed:

1. The authors focused on two SNPs, namely MKT189G and TAO34477C. A large number of *S. cerevisiae* genomes have been sequenced in recent years, with now more than 3,000 genomes available (PMID: 39559979). How common are these SNPs at the species level? It would be interesting to provide this information somewhere.
2. Depending on the results and the frequency in the population, it would be interesting to discuss the relevance.
3. The authors stated (page 5) 'while gene expression has been widely studied, its correlation levels is often limited'. This is true It's true but they forget to cite the latest, most exhaustive study on this subject (PMID: 38696467).

(Remarks on code availability)

Reviewer #2

(Remarks to the Author)

The current manuscript seems to be a continuation of the previous work done by the authors and aims to provide mechanistic insights into SNP-SNP interactions in dictating a complex phenotype. While this is an important field of study and has relevance for broader understanding for multiple SNPs interacting to impact biological processes, the manuscript in the current form falls short of expectations.

The authors have probably missed a big opportunity to provide deep-dive mechanisms and possibly reveal a 'systems-level' underpinning.

Major concerns:

1. Do MKT1 and TAO3 variants (in the S288c) background have lower expression/turnover? It is important to demonstrate if this is a functional change due to activity and/or dosage effect.

This will also help in interpreting the results including of the MmTt strain.

2. Fig-2C: It is often difficult to estimate consumption of 'nutrients' in the extracellular medium if the concentrations are in excess. The data is insufficient to claim 'no change' in nutrient uptake/metabolism. Flux measurement of differential intra-/extracellular acetate in addition to other paradigms should be used.

3. Fig-3: While it is temporal changes in gene expression is necessary to assess global transcriptional profile that is associated with the phenotype, the current measures do not offer any novel insights.

Since turnover rates of transcripts expressed at closely spaced time-points will likely be very low, such analyses do not have enough biological significance.

Moreover, GO analyses and all other further interpretations take into consideration 'probabilistic enrichment scores', such carryover effects may result in outputs with huge confounders.

It is best to capture actively transcribed mRNAs, using methods that are standard in the field, to provide clear unambiguous molecular basis.

In fact combining whole transcriptome with newly synthesized mRNAs will yield data that will unravel gene regulatory cascades, which are 'true' drivers of cumulative phenotypic outcome vis-a-vis sporulation.

It is also unclear the extent of overlap between SS and MMTT strains at each of the time points. Subtractive datasets should be provided, and for further bioinformatic analyses.

4. 3E: Continuous plotting and representation of oscillatory pattern are misleading since the periodicity of sampling is biased/non-uniform.

5. While enrichment of ribosomal assembly/synthesis and mitochondrial respiration is interesting, it is not surprising. Recent reports and emerging literature clearly demonstrate that these two pathways are pivotal to most biological processes and hence are enriched for a myriad of cellular functions. Therefore, it is unclear how this enrichment is any particular significance for sporulation in the context of MMTT. The current data does not give any further novel insights.

Ribo-tag sequencing will also aid in generating results to comment about protein allocation.

6. Any interpretation on allocation of proteome resources cannot be deduced simply based on changes in protein levels. Resource allocation for protein homeostasis is far more complex and involves energy expenditure for synthesis, folding and degradation.

Since the authors have most of the results from omics, they could assess cellular energy status and possibly derive mathematical correlates to define this better. Else this would again look speculative with no real biological meaning.

7. Protein allocation to meiotic cell cycle and carbohydrate metabolic processes sounds very vague. This could be a consequence of functional change rather than a cause and the consequential effect may simply arise from degradation of other proteins due to change in cellular state.

8. Fig-6: The additive phenotype w.r.t arg mutants is indeed interesting but the results are insufficient to support the conclusions. Further, the differential effects on mitochondrial respiration could be a secondary or tertiary effect of intermediary metabolism and likely a compensation. Also not all of the parameters scored show additivity pointing towards a complex interplay.

9. The authors have pitched the story as being important to understand general principles of gene-environment interplay. There is not much merit in that argument since acetate induced sporulation is not truly 'environmental' in the classical sense unless they have tried multiple other inducers. Also the manuscript reads verbose with very loose scientific justifications.

(Remarks on code availability)

As above.

Reviewer #3

(Remarks to the Author)

Overall a comprehensive study that provides new insights into yeast sporulation process by combining genetic and omics approaches. Below are my suggestions for improving the study.

1. The role of arginine pathway is unclear – could authors add discussion on how arginine biosynthesis links to respiration. Can the phenotype of the Mt strain be reversed by supplementing arginine?

2. Additive effect of MMTT (Fig 2) – please provide the numbers in the text since from the bar plots it seems that the effect is slightly less than the sum of the individual mutations.

3. "...better nutrient properties of the SNPs" -> incorrect phrasing. It's the properties of the strains harbouring the SNPs.
4. Fig 3C – please show the actual data points.
5. "Genetic interactions are prevalent across species": this sentence is unnecessary because it is rather obvious and direct consequence of the complexity of cellular organisation.
6. "However, the molecular mechanisms underlying these interactions remain largely unexplored" -> this is rather unhelpful contextualization since mechanism will be different for each interaction.
7. "novel unique pathways" -> unclear what this refers to since no new pathways are demonstrated in the study. The correct interpretation of the data is that the double mutant exhibits phenotype through different pathways than the corresponding single mutants.
8. More explanation should be provided for the proteomics data and why the authors believe that it can be used in a quantitative sense.
9. The link to human disease should be removed/toned down. Extrapolation even to different yeast strain backgrounds is tricky as authors know well, so any extrapolation to human case is far-fetched.

(Remarks on code availability)

Version 1:

Reviewer comments:

Reviewer #1

(Remarks to the Author)

The authors have addressed my comments satisfactorily. This is an excellent paper, and I recommend it for publication.

(Remarks on code availability)

Reviewer #2

(Remarks to the Author)

Minor edits to discussion to synthesize the results and narrative will help.

(Remarks on code availability)

Authors are encouraged to sharpen the manuscript and better integrate the new data from metabolomics to interpret the overall phenotype.

Reviewer #3

(Remarks to the Author)

I am satisfied with the response.

(Remarks on code availability)

RESPONSE TO REVIEWERS' COMMENTS

We thank the reviewers and editors for their valuable feedback. Below, we provide a point-by-point response to each comment. Reviewer comments are presented in **black**, our responses are in *italics*, and changes made to the manuscript are shown in **blue**.

REVIEWER 1

Genetic interactions, essential in shaping complex traits across species, are often poorly understood. This study investigates how interactions between two SNPs, MKT189G and TAO34477C, in *Saccharomyces cerevisiae* influence sporulation. By integrating gene expression and proteomics data, the authors reveal that these SNPs activate the arginine biosynthesis pathway while suppressing ribosome biogenesis under nutrient starvation, highlighting a metabolic trade-off. This SNP interaction enhances sporulation by modulating cellular stress responses. The study also connects these findings to human diseases like cancer, where similar SNP combinations may disrupt metabolic pathways, leading to uncontrolled cell proliferation.

This is interesting work leading to a more precise view of key aspects of genetic interaction at different levels. The question is clearly posed and the analyzes give interesting results. This study is important because it provides strong experimental data to support the understanding of one of the keys to the genotype-phenotype relationship. This is a well-conducted study.

Overall, this is very nice story with interesting conclusions.

We thank the reviewer for the positive feedback on our study and for appreciating its importance in understanding key changes in genotype-phenotype relationships. We are encouraged that the reviewer found our analyses and conclusions compelling.

I just have a couple of minor points that need to be addressed:

1. The authors focused on two SNPs, namely MKT189G and TAO34477C. A large number of *S. cerevisiae* genomes have been sequenced in recent years, with now more than 3,000 genomes available (PMID: 39559979). How common are these SNPs at the species level? It would be interesting to provide this information somewhere.

Thank you for this suggestion. We found that the MKT1(89G) is a common variant, while the TAO3(4477C) SNP is an ultra-rare allele. We have now incorporated this information into the result section (Page 5; Line no. 103- 108)

*From the comprehensive yeast genomics dataset of 3,034 strains⁴⁹, we found that the MKT1^{89G} SNP has an allele frequency of 0.9955, indicating that it is a common variant in *S. cerevisiae*. In contrast, the TAO3^{4477C} SNP has an allele frequency of 4.952×10^{-4} ,*

highlighting that it is an ultra-rare allele in the population. As these are independent SNPs, we expect around 0.0493% of the population to carry both alleles in combination.

2. Depending on the results and the frequency in the population, it would be interesting to discuss the relevance.

We have discussed the relevance in the discussion section, where we highlight how interactions between the common variant $MKT1^{89G}$ and a rare variant $TAO3^{4477C}$ modulate genetic and environment-specific interactions. We have also added the relevance of these findings in the context of human diseases (Page 20; Lines 555- 571).

The observed allele frequencies of $MKT1^{89G}$ and $TAO3^{4477C}$ indicate distinct evolutionary dynamics. The near-fixation of $MKT1^{89G}$ in *S. cerevisiae* suggests it confers an adaptive advantage, particularly in stress-related contexts, as shown in prior studies^{69,70}. In contrast, the $TAO3^{4477C}$ variant is extremely rare, implying it may be deleterious in most backgrounds or advantageous only under specific environmental or genetic conditions while having an effect size similar to a common variant⁷¹. Notably, when $TAO3^{4477C}$ co-occurs with $MKT1^{89G}$, it induces a metabolic shift activating arginine biosynthesis and suppressing ribosome biogenesis, indicating a context-dependent benefit that arises from this genetic interaction.

Beyond yeast, these findings provide a conceptual framework for understanding how genetic interactions (G×G) can reconfigure cellular metabolism to activate latent pathways. In human systems, genetic interactions further modified by environmental factors may activate latent pathways that can influence disease phenotypes or therapeutic responses, especially in conditions where metabolic reconfiguration and rare allele effects are observed. Our work emphasises the importance of resolving variant effects in combinatorial contexts and demonstrates how integrated multi-omics approaches can uncover the molecular logic of complex trait architecture.

3. The authors stated (page 5) ‘while gene expression has been widely studied, its correlation levels is often limited’. This is true It's true but they forget to cite the latest, most exhaustive study on this subject (PMID: 38696467).

Thanks for pointing it out. Now we have cited the relevant study in the main text (Page 10; Line 256).

REVIEWER 2

The current manuscript seems to be a continuation of the previous work done by the authors and aims to provide mechanistic insights into SNP-SNP interactions in dictating a complex phenotype. While this is an important field of study and has relevance for broader understanding for multiple SNPs interacting to impact biological processes, the manuscript in the current form falls short of expectations.

The authors have probably missed a big opportunity to provide deep-dive mechanisms and possibly reveal a 'systems-level' underpinning.

We appreciate the reviewer's extensive feedback and constructive criticism. We have carefully considered the comments and have addressed the major concerns by incorporating additional experiments and clarifying key points in the manuscript. We believe these revisions strengthen the mechanistic insights and bring us closer to revealing the systems-level basis of the observed SNP-SNP interactions.

Major concerns:

1. Do MKT1 and TAO3 variants (in the S288c) background have lower expression/turnover? It is important to demonstrate if this is a functional change due to activity and/or dosage effect.

This will also help in interpreting the results including of the MmTt strain.

Differential gene expression analysis for MKT1 variants in MM strain (Gupta et al., 2015) and TAO3 variants in TT strain (Gupta et al., 2016), and our RNASeq analysis of MKT1 and TAO3 variants in MMTT strain compared to SS (S288c) strain showed that there was no difference in the gene expression values of any variant allele in any of the backgrounds.

However, at the protein level, expression of MKT1 protein (MKT1^{89A}) was close to zero in both SS and TT strains. In the MM and MMTT strains, the MKT1 protein (MKT1^{89G}) was significantly expressed compared to the SS and TT strains. This showed that the protein formed in SS and TT strains has a low turnover rate when compared to MM and MMTT strains (Supplementary Fig. 8).

For TAO3, we did not observe any significant change in gene and protein levels across the SS, MM, TT, and MMTT strains. This suggests that the variants likely cause functional changes in the protein rather than changes in expression or turnover rate.

We have now included this information in the results section (Page 11; Lines 300-310)

We first verified whether MKT1 and TAO3 genes are differentially regulated at the protein level, given that no differences were observed at the gene expression level. Interestingly, we

found that *MKT1* protein levels were nearly undetectable in both the SS and TT strains, which carry the *MKT1*^{89A} variant. In contrast, *MKT1* protein was expressed in the MM and MMTT strains, both of which carry the *MKT1*^{89G} variant. This indicates that the *MKT1* protein encoded by the 89A variant likely has reduced stability or increased degradation compared to the 89G variant. For *TAO3*, we did not observe any significant changes in either gene or protein expression across the SS, MM, TT, and MMTT strains. This suggests that the *TAO3*^{4477C} variant may alter protein function rather than affecting expression levels or protein stability (Supplementary Fig. 8).

Supplementary Fig. 8: Protein expression levels of *MKT1* and *TAO3* across yeast strains and time points. Boxplots showing the mass fraction of the proteome (%) allocated to the *MKT1* (panels A, B) and *TAO3* (panels C, D) and in four yeast strains (SS, MM, TT, and MMTT) at two time points: 0 h (A, C) and 2 h 30 min (B, D) after sporulation induction. Statistical comparisons were performed using unpaired t-test between SS and each of the other strains (MM, TT, MMTT). Significance levels are indicated as follows: ns, not significant; *, p < 0.05; **, p < 0.01.

2. Fig-2C:

It is often difficult to estimate consumption of 'nutrients' in the extracellular medium if the concentrations are in excess. The data is insufficient to claim 'no change' in nutrient uptake/metabolism. Flux measurement of differential intra-/extracellular acetate in addition to other paradigms should be used.

We agree with the reviewer that assessing nutrient uptake and metabolic fluxes in a dynamic system like sporulation requires a multi-dimensional approach, especially when extracellular concentrations remain high and may mask subtle uptake differences. To address this concern, we have performed a series of complementary experiments that collectively strengthen the evidence for SNP-associated changes in acetate metabolism and nutrient utilisation:

1. Extracellular Acetate Profiling:

*We conducted a time-resolved analysis of extracellular acetate levels using HPLC. These results showed that the MMTT strain, which displayed the highest sporulation efficiency, also exhibited the most pronounced reduction in extracellular acetate over time. While all strains showed comparable acetate levels at 2 hours, significant differences emerged afterwards, with MMTT showing a steep decrease from 2 to 8 hours, suggesting more efficient uptake. The MM and TT strains also outperformed the SS strain in acetate utilisation during the later phases. We have added this experimental result in the Results section "**Role of MKT1^{89G} and TAO3^{4477C} SNPs in sporulation efficiency variation**" and added it as Figure 2C (Page 6; Lines 129-133).*

2. Intracellular Acetate Quantification:

To complement extracellular measurements, we quantified intracellular acetate using an enzymatic assay. The MMTT strain showed rapid depletion of intracellular acetate within the first 8 hours, followed by a gradual accumulation up to 24 hours. In contrast, other strains exhibited different temporal patterns where TT showed early fluctuations, and SS and MM displayed more conservative acetate dynamics. A gradual decrease in intracellular acetate concentration in the MMTT strain compared to other strains indicated that downstream metabolic pathways were consistently activated, supporting better sporulation efficiency. These differential extra- and intracellular acetate consumption rates indicate that SNPs influence both the timing and efficiency of acetate utilisation, supporting the hypothesis of altered metabolic regulation in different genetic backgrounds. These results have been added on Page 6, Lines 133-145 and Figure 2D.

Sporulation in yeast is induced when strains are grown in an acetate medium, the sole carbon source. We hypothesised that increased sporulation efficiency of MM, TT and MMTT strains could be due to altered acetate uptake, as previous studies have shown activation of key metabolic pathways like nitrogen metabolism, TCA cycle and gluconeogenesis associated with the SNPs involved^{40,41}. To test this, we measured extracellular acetate levels over time. MMTT strain showed a sharp decline in acetate compared to SS, MM, and TT strains, which were similar at 2 hours but diverged later (Fig. 2C). Further, we observed that the MM and TT strains outperformed SS in utilisation, particularly after 8 hours (Fig. 2C). Intracellular acetate analysis revealed rapid consumption in MMTT within 8 hours, followed by gradual accumulation up to 24 hours (Fig. 2D), suggesting activation of downstream metabolism during the early stages of sporulation. The TT strain showed a biphasic usage with peaks at 2 h 30 min and 8–12 hours. The SS and MM strains had similar trends with an early dip, followed by an accumulation till 12 hours and then a decline.

These observations suggested that the enhanced sporulation efficiency observed in the MMTT strain could be driven by more efficient acetate uptake and utilisation during the early timepoints (0–8 h). The distinct temporal patterns of intracellular and extracellular acetate across strains suggest that the SNP-associated changes can confer metabolic plasticity across strains. This plasticity is likely orchestrated by coordinated changes at the transcriptomic, proteomic, and intracellular metabolite levels, enabling differential sporulation efficiency across strains.

Figure 2C: Extracellular acetate levels in the sporulation medium of the SS, MM, TT, and MMTT strains were measured using HPLC at different time points during sporulation. Error bars represent the mean \pm SD of two independent biological replicates.

Figure 2D: Intracellular acetate concentrations were measured across four yeast strains (SS, MM, TT, MMTT) at five time points (0 h, 2 h30 min, 5 h40 min, 12 h, and 24 h) in sporulation medium. Error bars represent mean \pm SD from three biological replicates. Individual data points are shown as dots. Statistical significance was assessed using one-way ANOVA, followed by Tukey's HSD post-hoc test for each time point independently. **** $p < 1e^{-4}$, *** $p < 1e^{-3}$, ** $p < 1e^{-2}$, * $p < 5e^{-2}$. Source data are provided as a Source Data file for (B-D).

3. Targeted Amino Acid Profiling:

To further understand the metabolic consequences of altered acetate dynamics, we performed targeted metabolite profiling of 20 amino acids at 0h, 2h30m, and 8h during sporulation. The MMTT strain exhibited a unique biphasic amino acid trajectory, with an early surge in key amino acids (e.g., arginine, glutamate, methionine, valine), followed by rapid depletion, consistent with early biosynthetic activation and subsequent metabolic commitment. In contrast, the SS strain accumulated amino acids gradually, suggesting a more resource-conserving strategy. Notably, changes in arginine concentration in this profiling paralleled transcriptional and proteomic evidence, where distinct temporal control of amino acid pools, especially those linked to nitrogen metabolism, results in the activation of the arginine biosynthesis pathway as a key driver of enhanced sporulation in MMTT. These results have been added in Pages 15 & 16; Lines 399-435.

Together, these data provide a systems-level view of how SNP-SNP interactions rewire metabolic fluxes, particularly in acetate and nitrogen metabolism. The combination of extracellular, intracellular, and metabolite dynamics presented here offers strong support for SNP-associated metabolic plasticity. These coordinated changes at multiple omic layers collectively shape the observed differences in sporulation efficiency. We have modified the Results section "**Role of MKT1^{89G} and TAO3^{4477C} SNPs in sporulation**

efficiency variation” and added a new results section “Genetic interactions alter intracellular amino acid dynamics linked to nitrogen metabolism”. The raw intensity files are given as Supplementary Data 9.

Genetic interactions alter intracellular amino acid dynamics linked to nitrogen metabolism

Through our transcriptomics and proteomics data, we revealed a distinct regulation of amino acid metabolism in the MMTT strain. Hence, we wanted to test and show how the genetic interactions can reshape metabolic trajectories during sporulation. For this, we performed targeted temporal profiling of key amino acids across the SS, MM, TT, and MMTT strains at key developmental time points (0 h, 2 h 30 min, and 8 h) of sporulation (Fig. 6A).

Normalised intensity values were averaged across replicates and represented as z-score heatmaps. Clustering was performed on the MMTT strain to highlight its metabolic program, and the same amino acid order was retained across all strains for comparative analysis (Supplementary Fig. 14). We found that the MMTT strain showed a unique biphasic amino acid trend, which was not observed in the other strains (Fig. 6B; Supplementary Fig. 14). Notably, there was an early surge in the levels of alanine, arginine, lysine, glutamine, and histidine between 0 h - 2 h 30 min, followed by a pronounced depletion phase from 2 h 30 min-8 h (Fig. 6B). This trajectory was consistent with our transcriptomic data, where we found a unique upregulation of genes related to amino acid metabolism, particularly histidine metabolism, arginine biosynthesis during the early phase (Supplementary Data 3). This shows an early biosynthetic burst, possibly driven by a programmed metabolic activation, followed by rapid mobilisation of these resources for nucleotide biosynthesis during commitment to meiosis and spore morphogenesis. In contrast, the SS strain maintained a continuous accumulation of most amino acids, suggesting a dysregulation of intracellular metabolite utilisation for meiosis and sporulation (Fig. 6C).

Additionally, glutamic acid, an early intermediate derived from acetate via the TCA cycle, was found to show a distinct early spike exclusively in the MMTT strain, followed by sustained levels up to 8 hours (Fig. 6C). This pattern was absent in the other strains. The early accumulation of glutamate in the MMTT strain was consistent with previous studies reporting a transient increase in glutamate during early sporulation, which facilitated ammonium ion removal, a known inhibitor of the sporulation process⁵⁶.

Together, these data showed that amino acid metabolism was not merely passive during sporulation but was actively rewired in a strain-specific manner. The distinct temporal control of amino acid pools, especially those linked to nitrogen metabolism,

emerged as a critical determinant of sporulation trajectory and efficiency, with the MMTT strain showing a dynamic and resource-intensive strategy to support its developmental program. Raw mass spectrometry intensity data for all measured amino acids are provided in Supplementary Data 9.

Supplementary Fig. 14: Heatmap showing the z-scores of amino acid intensity values across time points and strains. Amino acids are ordered based on hierarchical clustering (with K=8) of z-score values in the MMTT strain, keeping time as a constant.

3. Fig-3: While it is temporal changes in gene expression is necessary to assess global transcriptional profile that is associated with the phenotype, the current measures do not offer any novel insights.

Since turnover rates of transcripts expressed at closely spaced time-points will likely be very low, such analyses do not have enough biological significance.

Moreover, GO analyses and all other further interpretations take into consideration 'probabilistic enrichment scores', such carryover effects may result in outputs with huge confounders.

It is best to capture actively transcribed mRNAs, using methods that are standard in the field, to provide clear unambiguous molecular basis.

In fact combining whole transcriptome with newly synthesized mRNAs will yield data that will unravel gene regulatory cascades, which are 'true' drivers of cumulative phenotypic outcome vis-a-vis sporulation.

We thank the reviewer for this insightful comment. We agree that post-transcriptional regulation, including RNA turnover and nascent transcription, can provide valuable mechanistic insights, especially in linking transcriptional changes to proteomic outcomes. However, the main objective of our study was to characterise the global transcriptional dynamics during the early stages of sporulation, with a focus on identifying differentially expressed genes that may underlie phenotypic divergence between yeast strains.

Our temporal design was specifically aimed at identifying early, potentially causal transcriptional events that differentiate strains, rather than dissecting RNA stability or nascent transcription. While integrating nascent and total RNA data could certainly enhance mechanistic resolution and reveal transcriptional cascades, such analyses represent a valuable future direction rather than a requirement for addressing our current research question. Our approach, which involves densely sampled time points, is consistent with prior work in yeast that has successfully used whole transcriptome and proteome data to get biological insights (Weiner et al., 2012; Gupta et al., 2015, 2016; Wen et al., 2016). Further, the effect of these alleles results in the initiation of spore formation in MMTT and SK1 (original parent of these alleles) by 8 h.

To address some of the confounding effects and to strengthen the link between gene expression and phenotype, we have now correlated the gene expression changes with the intracellular levels of key amino acids across three time points during sporulation in all four strains, as discussed in the previous section (Page 15; Line 412-422).

Notably, there was an early surge in the levels of alanine, arginine, lysine, glutamine, and histidine between 0 h - 2 h 30 min, followed by a pronounced depletion phase from 2 h 30 min-8 h (Fig. 6B). This trajectory was consistent with our transcriptomic data, where we found a unique upregulation of genes related to amino acid metabolism, particularly histidine metabolism, arginine biosynthesis during the early phase (Supplementary Data 3). This shows an early biosynthetic burst, possibly driven by a programmed metabolic activation, followed by rapid mobilisation of these resources for nucleotide biosynthesis during commitment to meiosis and spore

morphogenesis. In contrast, the SS strain maintained a continuous accumulation of most amino acids, suggesting a dysregulation of intracellular metabolite utilisation for meiosis and sporulation (Fig. 6C).

These data show strain-specific trends, particularly in the MMTT strain, which exhibited elevated amino acid levels up to 2 h 30 min, followed by a marked reduction consistent with active utilisation. These metabolic trends support our transcriptomic and proteomic findings, reinforcing the biological relevance of our observations.

It is also unclear the extent of overlap between SS and MMTT strains at each of the time points. Subtractive datasets should be provided, and for further bioinformatic analyses.

Thank you for the suggestion. We have now performed the analysis and incorporated the results. This observation has been included in the results section under the heading: “Temporal reshuffling of amino acid metabolism and ribosomal pathways in the presence of MKT1^{89G} and TAO3^{4477C} combination” (Page 8; Lines 192-199).

We also compared differential gene expression at each time point between the SS and MMTT strains. As expected, we identified a large number of differentially expressed genes in the MMTT strain, particularly at 2 h 30 min and 8 h 30 min (Supplementary Fig. 2, Supplementary Data 3). During the early stages of MMTT strain, upregulated genes were significantly enriched in pathways related to amino acid biosynthesis, including histidine metabolism and L-arginine biosynthetic processes. In the later stages, we observed enrichment of genes involved in the meiotic cell cycle and meiosis only in the MMTT strain (Supplementary Fig. 3).

Supplementary Fig.2: The number of differentially expressed genes in the MMTT strain when compared to the SS strain at each time point during sporulation.

Supplementary Fig.3: GO enrichment analysis for upregulated genes in the MMTT strain across each time point in comparison with the SS strain. The heatmap represents the p-value of each GO term.

4. 3E: Continuous plotting and representation of oscillatory pattern are misleading since the periodicity of sampling is biased/non-uniform.

Thank you for raising this important point. Below, we clarify our methodology. Firstly, we performed this non-uniform sampling to capture early transcriptional changes, which was in line with the previous studies, which showed that both MKT1-89G and TAO3-4477C had an early causal role in sporulation efficiency. Secondly, the Dirichlet Process Gaussian Process (DPGP) model does not assume uniform sampling intervals. It uses Gaussian processes to interpolate trajectories between observed time points (using the hyperparameter ‘true_times’), inherently accounting for irregular temporal spacing. This allows biologically meaningful clustering even with sparse or unevenly sampled data. We have now clarified this in the results section.

In brief, DPGP clusters the data using the Dirichlet process while modelling the temporal dependencies with Gaussian processes for non-uniform timepoints (Pages 8; Lines 213-215).

And in the Method section: Moreover, we utilised the `--true_times` parameter to preserve the temporal information associated with each expression profile during the clustering process due to non-uniform sampling timepoints in our data. This enabled us to capture the temporal dynamics inherent in the data accurately (Page 26; Lines 747--750).

5. While enrichment of ribosomal assembly/synthesis and mitochondrial respiration is interesting, it is not surprising. Recent reports and emerging literature clearly demonstrate that these two pathways are pivotal to most biological processes and hence are enriched for a myriad of cellular functions. Therefore, it is unclear how this enrichment is any particular significance for sporulation in the context of MMTT. The current data does not give any further novel insights.

We appreciate the reviewer's comment and agree that both ribosomal synthesis and mitochondrial respiration are fundamental cellular processes enriched in many biological contexts. Notably, several studies have elucidated how cells reallocate their proteome in response to changes in carbon sources and various nutrient starvation conditions. These studies, primarily employing proteome-constrained genome-scale models and quantitative proteomics, have consistently demonstrated how ribosomal pathways, mitochondrial functions and amino acid biosynthetic pathways are dysregulated under respiratory growth and nutrient limitation. This confirms the central role of these pathways in adapting to altered metabolic states (Bjorkeroth et al., 2020; Xia et al., 2022; Bartolomeo et al., 2020; Malina et al., 2021; Elsemman et al., 2022)

However, our study moves beyond general metabolic adaptation and focuses on the dynamic and variant-specific reallocation of proteome mass in response to SNP-SNP interactions during the initiation of sporulation. Here we rather than attributing proteome shifts solely to environmental triggers, we reveal how specific genetic interactions can reprogram core proteome allocation patterns, particularly enhancing mitochondrial and amino acid biosynthetic functions, highlighting a previously underappreciated layer of genetic control in metabolic regulation during early meiosis.

So it is that MMTT's improved respiratory and sporulation potential is due to the presence of $MKT1^{89G}$ and $TAO3^{4477C}$ alleles and the genetic interaction between them. This genetic interaction uniquely leads to (1) the downregulation of multiple ribosomal proteins uniquely in MMTT at the early sporulation time point and (2) a corresponding increase in proteome mass fraction allocated to mitochondrial translation, respiratory complex assembly, and arginine biosynthesis. These observations suggest a developmental trade-off: reduced investment in growth (ribosome production) in favour of metabolic readiness for sporulation, a resource shift that was not universal but uniquely emerges from the interaction of the causal $MKT1^{89G}$ and $TAO3^{4477C}$ alleles.

This shift in energy allocation was not observed in any of the other three strains, SS, MM, and TT.

This unique redirection of energy from the energy-rich process of ribosomal biosynthesis to amino acid biosynthesis in starvation conditions like sporulation is opposite to the observation made by Björkeröth et al. (2020), where the authors showed that in rich growth conditions, energy is allocated from amino acid biosynthesis to ribosomal biosynthesis.

Ribo-tag sequencing will also aid in generating results to comment about protein allocation.

To quantify proteome allocation changes, we employed absolute label-free proteomics using Data Independent Acquisition (DIA) and the Total Protein Approach (TPA), enabling robust estimation of protein concentrations (fmol/μg) across conditions. This method allowed direct comparison of proteome allocation across strains and time points, a strategy widely validated in studies of growth and nutrient-dependent adaptation in yeast. While ribosome profiling or RiboTag-seq can provide differential translational activity data, our quantitative proteomics approach offers complementary insights by capturing actual proteome investment. Together, our findings provide a systems-level view of how causal variants in combination reprogram proteome resource distribution to promote sporulation, offering mechanistic insight into genotype-to-phenotype relationships beyond pathway enrichment alone. (Page 10 ; Lines 266-273)

Employing label-free absolute quantitative proteomics using data-independent acquisition and Total Protein Approach (TPA approach), we quantified protein concentrations (in fmol/μg) at 2 time points, once at an initial time point (0 h) and during the early phase of sporulation (2 h 30 min). This approach provided a direct and quantitative view of proteome composition across conditions and genotypes. The absolute quantification enabled biologically meaningful comparisons of protein mass fractions, which were critical for interpreting proteome allocation and cellular resource distribution as described in previous studies^{20,53}.

6. Any interpretation on allocation of proteome resources cannot be deduced simply based on changes in protein levels. Resource allocation for protein homeostasis is far more complex and involves energy expenditure for synthesis, folding and degradation.

*We agree with the reviewer that interpreting proteome resource allocation involves more than just measuring protein abundance, as it also includes costs associated with synthesis, folding, degradation, and energy expenditure. Our approach was inspired by previous studies in *E. coli* and *S. cerevisiae* that have demonstrated how shifts in proteome fractions correlate with physiological states like specific growth rates or*

nutrient limitations (O'Brien et al., 2016; Lahtvee et al., 2017; Chen & Nielsen, 2021; Bjorkeroth et al., 2020; Xia et al., 2022, Bartolomeo et al., 2020). Below are a couple of studies that highlight the use of quantitative proteomics to infer proteome allocations in yeast.

1) Bjorkeroth et al. (2020) (<https://doi.org/10.1073/pnas.192189011>) did quantitative proteomics and found protein allocation patterns by summing the mass fraction of each protein belonging to a functional group, and found that the yeast strains allocate more proteome to translation and ribosomes from amino acid biosynthesis to grow faster in rich media.

2) Xia et al. (2022) (<https://doi.org/10.1038/s41467-022-30513-2>) systematically profiled proteome allocation across growth rates and nutrient limitations in yeast, revealing a linear correlation between most functional protein groups and specific growth rates under glucose limitation.

While we acknowledge that a complete energy budget for protein turnover is not captured in our dataset, the absolute proteomic quantification provides a critical and biologically meaningful readout of resource allocation trends. These trends are especially relevant in developmental programs like sporulation, where resource prioritisation, rather than growth optimisation, becomes the dominant strategy. Thus, this provides a first-order approximation of how proteome resources are being reallocated during critical transitions, such as sporulation.

We have now added this limitation to the Discussion section (Page 19; Lines 532-539).

While we acknowledge that a complete energy budget for protein turnover is not captured in our dataset, the absolute proteomic quantification provides a critical and biologically meaningful readout of resource allocation trends. These trends are especially relevant in developmental programs like sporulation, where resource prioritisation, rather than growth optimisation, becomes the dominant strategy. Thus, our protein allocation analysis provides a first-order approximation of how proteome resources are being reallocated during critical transitions, such as sporulation.

Since the authors have most of the results from omics, they could assess cellular energy status and possibly derive mathematical correlates to define this better. Else this would again look speculative with no real biological meaning.

Thank you for pointing this out. We have now expanded our analysis to integrate omics data with experimental measurements to better characterise the energy status of each strain and provide mechanistic insights grounded in biology rather than speculation.

To assess cellular energetics, we first analysed intracellular ATP levels over time. As shown in the revised results section (Page 14; Lines 382-387; Supplementary Fig. 12),

all strains exhibited a sharp drop in ATP levels until 2h30m. However, the MMTT strain showed complete ATP recovery by 8 hours, while the SS strain exhibited partial recovery, indicating differences in ATP regeneration capacity. The higher ATP regeneration capacity might be attributed to the efficient mitochondrial respiration in the MMTT strain

To investigate the underlying mechanisms, we utilised proteomic data to quantify protein allocation to energy metabolism pathways. Notably, the SS strain retained a high allocation to glycolysis (~14% of the proteome at 2h30m), whereas MM, TT, and MMTT showed reduced allocation (~11–12%) at the same time point (Supplementary Fig. 11). This suggests enhanced glycolytic capacity in SS, potentially contributing to substrate-level ATP regeneration even under limited mitochondrial activity. Consistently, Seahorse XF respirometry data showed significantly higher basal OCR in MM and MMTT compared to SS (now reported in the Results, Page 14 &15 ; Lines 387-398; Supplementary Fig. 13), consistent with increased proteomic investment in mitochondrial respiration pathways. These data support our interpretation that MMTT regenerates ATP primarily via oxidative phosphorylation, while SS relies more on glycolysis. Additionally, we examined intracellular glutamate levels, a proxy for TCA cycle activity, based on previous studies (Jambhekar and Amon, 2008; Ray and Ye, 2013). We observed that glutamate peaks at 2h30m in MMTT, suggesting elevated TCA cycle flux. This trend was less pronounced in SS, supporting lower mitochondrial metabolic activity.

We thank the reviewer for asking for this additional experimental data through which we were able to integrate our omics data with ATP levels, basal OCR, and glutamate dynamics to provide a biologically meaningful and evidence-based assessment of cellular energetics in our system.

Further, to validate our findings, we assessed the cellular energy status by analysing the intracellular ATP levels for all strains during sporulation at multiple timepoints. We observed that all the strains had a sharp decline in ATP levels by 2 h 30 min with the MMTT strain showing a complete recovery of the initial ATP concentration during 8 h, while other strains showed partial recovery (Supplementary Fig. 12). This enhanced recovery highlights the more efficient respiration during sporulation in the MMTT strain. Further to understand the increased ATP levels seen in the SS strain, we analysed the protein allocated to the glycolytic pathway, a key pathway for ATP production. At 2 h 30 min, the SS strain maintained a higher proportion of its proteome allocated to glycolytic enzymes (~14%), while MM, TT, and MMTT strains showed reduced allocation (~11–12%, Supplementary Fig. 11). This suggested that the SS strain retained a stronger glycolytic capacity, potentially supporting ATP regeneration through substrate-level phosphorylation even under conditions of limited mitochondrial activity. Consistently, Seahorse XF respirometry measurements revealed significantly

higher basal oxygen consumption rates (OCR) in MM and MMTT strains compared to SS (Supplementary Fig. 13), aligning with the proteomic evidence of increased investment in mitochondrial respiration.

Supplementary Fig. 12: Intracellular ATP concentrations over time in sporulation medium. ATP levels were quantified in strains SS, MM, TT and MMTT at 0 h, 1.16 h, 2.5 h, and 8 h. Data are presented as mean \pm SD from 2 to 3 biological replicates. Statistical significance was assessed using unpaired two-tailed t-tests for M, T and MT in comparison with the S strain (* $p < 0.05$, ** $p < 0.01$, *** $p < 0.001$; ns, not significant) ($n = 2$ or 3).

Supplementary Fig. 13: Basal oxygen consumption rate (OCR) across yeast strains. Boxplots show the normalised basal OCR (pmol/min per 0.1 OD) for each strain: SS, MM, TT, and MMTT. Each point represents an individual three Seahorse measurement under acetate medium conditions. Statistical comparisons were performed using pairwise t-tests with Bonferroni correction. Significant differences between groups are marked with asterisks (* $p < 0.05$, ** $p < 0.01$, *** $p < 0.001$, **** $p < 0.0001$).

7. Protein allocation to meiotic cell cycle and carbohydrate metabolic processes sounds very vague. This could be a consequence of functional change rather than a cause and the consequential effect may simply arise from degradation of other proteins due to change in cellular state.

We agree, and we have now removed this observation from the main manuscript.

8. Fig-6: The additive phenotype w.r.t arg mutants is indeed interesting but the results are insufficient to support the conclusions. Further, the differential effects on mitochondrial respiration could be a secondary or tertiary effect of intermediary metabolism and likely a compensation. Also not all of the parameters scored show additivity pointing towards a complex interplay.

We thank the reviewer for this thoughtful comment. We agree that the additive phenotype observed in the arg mutants is complex. Since not all parameters or pathways were identified or tested, and therefore, it is not possible for us to comment on their direct role in phenotypic additivity. This likely reflects a multifaceted interplay between mitochondrial function, nitrogen metabolism, and broader metabolic

compensation mechanisms. However, the convergence of metabolic, genetic, and phenotypic data in MMTT provides a compelling, multi-omic validation of our interpretation of the role of gene-gene interactions in activating latent metabolic pathways as a mechanism of phenotypic change, which leads to phenotypic additivity for sporulation efficiency phenotype. We have edited the section accordingly (Page 19; Lines 530-554).

9. The authors have pitched the story as being important to understand general principles of gene-environment interplay. There is not much merit in that argument since acetate induced sporulation is not truly 'environmental' in the classical sense unless they have tried multiple other inducers. Also the manuscript reads verbose with very loose scientific justifications.

To clarify, our primary objective was not to define environmental responses that affect sporulation, but rather to explore how, by studying intermediate molecular phenotypes such as mRNA and protein level changes, and amino acid dynamics, in a model system and a well-studied phenotype, like sporulation. We show that with the initial external environment (1% potassium acetate) held constant, the strains with different causal variants can display differential capacities to utilise the carbon source and exhibit distinct molecular trajectories during the sporulation process. These intrinsic, strain-specific responses reflect the interplay between genotype and the imposed environmental condition, revealing how different genetic architectures modulate a common environmental signal. We have now clarified this in the Discussion section (Page 18; Lines 514-518)

Further, potassium acetate is a well-established and physiologically relevant inducer of yeast sporulation widely used in laboratory settings to mimic extreme nutrient limitation, a natural environmental trigger for sporulation (Deutschbauer et al. 2002; Deutschbauer and Davis, 2005; Gerke et al., 2006, 2009, 2010; Sudarsanam and Cohen 2014; Tomar et al. 2013). Additionally, these two causal variants, MKT1^{89G} and TAO3^{4477C}, were identified as causal in SK1 x S288c segregant population phenotyped for sporulation efficiency in potassium acetate (Deutschbauer and Davis, 2005); we chose to perform our experiments in the same conditions. If we had changed the sporulation media, one would have to redo the whole segregant phenotyping and mapping causal variants, which was not the purpose of the manuscript.

We acknowledge the feedback regarding verbosity and have revised the manuscript for clarity, while strengthening the scientific rationale for our conclusions. We hope the revised version is concise and better communicates the focus and significance of our findings.

Complex trait variation arises from both genetic and environmental influences. When multiple causal genes contribute to a trait, the resulting phenotype can be shaped by gene–gene–environment (G×G×E) interactions. To isolate the role of gene–gene (G×G) interactions, we held the environment constant and examined the consequences of genetic interactions to the phenotypic variation by studying the intermediate phenotypes like gene expression, protein expression and metabolite levels

Reviewer #2 (Remarks on code availability):

As above.

All the codes and data are available either as Supplementary Information or as data files deposited in public repositories.

REVIEWER 3

Overall a comprehensive study that provides new insights into yeast sporulation process by combining genetic and omics approaches. Below are my suggestions for improving the study.

We are encouraged that the reviewer found our analyses and conclusions convincing. We have addressed the comments and revised the manuscript accordingly, including adding new discussion points and refining overstatements related to human diseases.

1. The role of arginine pathway is unclear – could authors add discussion on how arginine biosynthesis links to respiration. Can the phenotype of the Mt strain be reversed by supplementing arginine?

To test if arginine supplementation can rescue the growth and sporulation defect MMTT- Δ arg4 strain, we supplemented YPD and potassium acetate media with the following amino acids: arginine, glutamate, glutamine, and leucine added individually. Interestingly, none of these amino acid supplementations rescued the growth (see Supplementary Fig. 15) and sporulation defects as observed in a previous study (Campero-Basaldua et al, 2023) and Page 17; Lines: 470-472 from the Results section). This relationship between the arginine biosynthetic pathway and mitochondrial function has been added in the discussion section (Page 19; Lines 539-554 from Discussion).

Further, through experimental validation, we have shown that ARG4 is necessary to maintain mitochondrial activity and respiration only when MKT1^{89G} and TAO3^{4477C} SNPs are combined. We speculate that due to the genetic interactions between MKT1^{89G} and TAO3^{4477C} SNPs, a rewiring of the metabolic network makes mitochondrial function heavily dependent on the arginine biosynthetic pathway. This was further supported by the unique biphasic dynamics of alanine and arginine in the MMTT strain, which accumulated early (0 h - 2 h 30 min) and was rapidly depleted by 5h 40 min, unlike in other strains. The interaction observed here shares conceptual parallels with known “moonlighting” functions of metabolic enzymes. For example, Alt1, an alanine transaminase, has been implicated in mitochondrial gene regulation beyond its catalytic activity⁶⁶. Similarly, Ilv5, involved in branched-chain amino acid biosynthesis, also contributes to mtDNA stability⁶⁷. These findings indicate that metabolic enzymes, including those in the arginine pathway, might participate in mitochondrial regulation beyond their canonical roles, particularly under conditions of network rewiring imposed by genetic variation⁶⁸.

Supplementary Fig. 15: Amino acid supplementation does not rescue the growth defect of MMTT-arg4Δ. Bar plot showing the growth rates of the MMTT-arg4Δ strain in YPD and YPD supplemented with individual amino acids. Growth rates were measured under each condition, and the mean ± SD is shown. Statistical significance was assessed using an unpaired two-sided t-test comparing each supplemented condition to the YPD control. Significance levels are indicated as follows: ns, not significant; *, p < 0.05; **, p < 0.01.

2. Additive effect of MMTT (Fig 2) – please provide the numbers in the text since from the bar plots it seems that the effect is slightly less than the sum of the individual mutations.

As suggested by the reviewer, we have now added the numbers to the main results (Page 5; Lines 116-120) which clearly indicate increased sporulation efficiency in MMTT strain after 48 hours in sporulation medium compared to the S288c strain.

3. “...better nutrient properties of the SNPs” -> incorrect phrasing. It’s the properties of the strains harbouring the SNPs.

We have now changed it. Thanks for pointing it out

4. Fig 3C – please show the actual data points.

While it is difficult to add these numbers to the heatmap, we have added this to the source data file, which accompanies the manuscript, where all the p-values of heatmaps in Fig. 3 are given.

5. “Genetic interactions are prevalent across species”: this sentence is unnecessary because it is rather obvious and direct consequence of the complexity of cellular organisation.

We have now removed this line from the manuscript and added “Genetic interactions are fundamental to the architecture of complex traits” (Page 2, Line 14)

6. “However, the molecular mechanisms underlying these interactions remain largely unexplored” -> this is rather unhelpful contextualization since mechanism will be different for each interaction.

We have now edited this line from the manuscript and added “Genetic interactions are fundamental to the architecture of complex traits, yet the molecular mechanisms by which variant combinations influence cellular pathways remain poorly understood” (Page 2, Line 14)

7. “novel unique pathways” -> unclear what this refers to since no new pathways are demonstrated in the study. The correct interpretation of the data is that the double mutant exhibits phenotype through different pathways than the corresponding single mutants.

Thanks for pointing it out. We have now changed the novel pathways to unique latent pathways (Page 3; Line 57, Page 4; Line 88 and Page 20; Line 567).

8. More explanation should be provided for the proteomics data and why the authors believe that it can be used in a quantitative sense.

Thank you for this comment. We employed absolute quantitative proteomics using a label-free, Data Independent Acquisition (DIA) strategy combined with the Total Protein Approach (TPA), which enabled estimation of protein concentrations (in fmol/μg) without relying on external standards (<https://orbit.dtu.dk/en/publications/6ced90b7-de9c-4445-bb8d-c0a58b4959c8>). This approach provided a direct and quantitative view of proteome composition across conditions and genotypes. The absolute quantification enabled biologically meaningful comparisons of protein mass fractions, which were critical for interpreting proteome allocation and cellular resource distribution. Our results align with previous studies that utilised absolute proteomics to uncover proteome allocation shifts in yeast under nutrient limitations and varying growth conditions (e.g., MetzI-Raz et al., 2017; Bjørn et al., 2021; Xia et al, 2022; Bjorkeroth et al., 2020), validating the robustness of our approach. We have now added this to our main manuscript (Page 10; Lines : 266-2723

Employing label-free absolute quantitative proteomics using data-independent acquisition and Total Protein Approach (TPA approach), we quantified protein concentrations (in fmol/μg) at 2 time points, once at an initial time point (0 h) and

during the early phase of sporulation (2 h 30 min). This approach provided a direct and quantitative view of proteome composition across conditions and genotypes. The absolute quantification enabled biologically meaningful comparisons of protein mass fractions, which were critical for interpreting proteome allocation and cellular resource distribution as described in previous studies^{20,53}

9. The link to human disease should be removed/toned down. Extrapolation even to different yeast strain backgrounds is tricky as authors know well, so any extrapolation to human case is far-fetched.

We agree that it is important to find these latent pathways for each of the genetic interactions and have now toned down and rewritten the discussion section (Page 20; Lines 564-571). However, the concept of latent pathways being activated by genetic interaction has been highlighted.

Beyond yeast, these findings provide a conceptual framework for understanding how genetic interactions (G×G) can reconfigure cellular metabolism to activate latent pathways. In human systems, genetic interactions further modified by environmental factors may activate latent pathways that can influence disease phenotypes or therapeutic responses, especially in conditions where metabolic reconfiguration and rare allele effects are observed. Our work emphasises the importance of resolving variant effects in combinatorial contexts and demonstrates how integrated multi-omics approaches can uncover the molecular logic of complex trait architecture.

References cited in the response:

1. Lahtvee, P.-J. et al. Absolute Quantification of Protein and mRNA Abundances Demonstrate Variability in Gene-Specific Translation Efficiency in Yeast. *Cell Systems* **4**, 495-504.e5 (2017).
2. Chen, Y. et al. Reconstruction, simulation and analysis of enzyme-constrained metabolic models using GECKO Toolbox 3.0. *Nat Protoc* 1–39 (2024) doi:10.1038/s41596-023-00931-7.
3. Björkeröth, J. et al. Proteome reallocation from amino acid biosynthesis to ribosomes enables yeast to grow faster in rich media. *Proceedings of the National Academy of Sciences* **117**, 21804–21812 (2020).
4. Xia, J. et al. Proteome allocations change linearly with the specific growth rate of *Saccharomyces cerevisiae* under glucose limitation. *Nat Commun* **13**, 2819 (2022).
5. Di Bartolomeo, F. et al. Absolute yeast mitochondrial proteome quantification reveals trade-off between biosynthesis and energy generation during diauxic shift. *Proceedings of the National Academy of Sciences* **117**, 7524–7535 (2020).
6. Deutschbauer, A. M. & Davis, R. W. Quantitative trait loci mapped to single-nucleotide resolution in yeast. *Nature Genetics* **37**, 1333–1340 (2005).
7. Deutschbauer, A. M., Williams, R. M., Chu, A. M. & Davis, R. W. Parallel phenotypic analysis of sporulation and postgermination growth in *Saccharomyces cerevisiae*. *Proceedings of the National Academy of Sciences of the United States of America* **99**, 15530–15535 (2002).
8. Gerke, J. P., Chen, C. T. L. & Cohen, B. A. Natural isolates of *Saccharomyces cerevisiae* display complex genetic variation in sporulation efficiency. *Genetics* **174**, 985–997 (2006).
9. Gerke, J., Lorenz, K., Ramnarine, S. & Cohen, B. Gene-environment interactions at nucleotide resolution. *PLoS Genetics* **6**, (2010).
10. Sudarsanam, P. & Cohen, B. A. Single Nucleotide Variants in Transcription Factors Associate More Tightly with Phenotype than with Gene Expression. *PLoS Genetics* **10**, (2014).
11. Gupta, S. et al. Meiotic interactors of a mitotic gene *Tao3* revealed by functional analysis of its rare variant. *G3: Genes, Genomes, Genetics* **6**, 2255–2263 (2016).
12. Gupta, S. et al. Temporal Expression Profiling Identifies Pathways Mediating Effect of Causal Variant on Phenotype. *PLoS Genetics* **11**, 1–23 (2015).

13. Teyssonnière, E. M. et al. *Species-wide quantitative transcriptomes and proteomes reveal distinct genetic control of gene expression variation in yeast. Proceedings of the National Academy of Sciences* **121**, e2319211121 (2024).

REVIEWERS' COMMENTS

Reviewer #1 (Remarks to the Author):

The authors have addressed my comments satisfactorily. This is an excellent paper, and I recommend it for publication.

We sincerely thank Reviewer #1 for the positive evaluation and recommendation for publication. We are glad that the revisions addressed your previous concerns satisfactorily.

Reviewer #2 (Remarks to the Author):

Minor edits to discussion to synthesize the results and narrative will help. Authors are encouraged to sharpen the manuscript and better integrate the new data from metabolomics to interpret the overall phenotype.

We thank the reviewer for accepting our revisions and giving positive feedback. We have added a few lines on how the observed pattern seen in MMTT strain will be helpful for contributing to high sporulation efficiency.

We have carefully looked for amino acids that show early surge and are followed by utilization patterns for better interpretation, as amino acid metabolism was differentially regulated at both transcriptome and proteome levels. By doing this, we were able to observe key amino acids related to nitrogen metabolism showing biphasic trends only in MMTT strain, validating our transcriptomics and proteomics data.

We have added the following line in the Discussion section (lines 545-548):

The increased sporulation efficiency in the MMTT strain could be attributed to the immediate response to nutrient starvation and their ability to synthesize the necessary amino acids like histidine during the early phase as precursors for nucleotide biosynthesis and meiotic process during the later stages.

Reviewer #3 (Remarks to the Author):

I am satisfied with the response.

We thank Reviewer #3 for the positive feedback and are pleased to know that the revisions addressed the concerns raised earlier.